# ReFeed: Multi-dimensional Summarization Refinement with Reflective Reasoning on Feedback

**Taewon Yun**[1], **Jihwan Oh**[1], **Hyangsuk Min**[1]
**Yuho Lee**[1], **Jihwan Bang**[1], **Jacon Cai**[2]*, **Hwanjun Song**[1†]
[1]Korea Advanced Institute of Science and Technology (KAIST)
[2]Amazon Web Services, AI Labs
{ytaewon0415, songhwanjun}@kaist.ac.kr

## Abstract

Summarization refinement faces challenges when extending to multi-dimension. In this paper, we introduce ReFeed, a powerful summarization refinement pipeline that enhances multiple dimensions through reflective reasoning on feedback. To achieve this, we release SumFeed-CoT, a large-scale Long-CoT-based dataset optimized for training a lightweight model with reflective reasoning. Our experiments reveal how the number of dimensions, feedback exposure, and reasoning policy influence refinement performance, highlighting reflective reasoning and simultaneously addressing multiple feedback is crucial to mitigate trade-off between dimensions. Furthermore, ReFeed is robust to noisy feedback and feedback order. Lastly, our finding emphasizes that creating data with a proper goal and guideline constitutes a fundamental pillar of effective reasoning. The dataset and model is available at https://github.com/DISL-Lab/ReFeed.

## 1 Introduction

In text summarization, large language models (LLMs) have primarily aimed to enhance faithfulness by incorporating advanced optimization methodologies, such as SYNFAC-EDIT (Mishra et al., 2024) and CPO (Feng et al., 2024). However, this emphasis on a single dimension, such as faithfulness, raises an important question: *"Does improving only one dimension come at the cost of other aspects of summarization?"* Generally, aligning model's responses with a single aspect often compromises other crucial aspects of quality (Guo et al., 2024; Song et al., 2025). In light of these conflicts, it becomes important to enhance LLM's responses from a multi-dimensional perspective. Yet, effectively improving all dimensions remains challenging not only because collecting multifaceted data is costly (Song et al., 2025) but also optimizing multi-dimension is complex (Ryu et al., 2024).

One potential pathway is to refine an initial response through *post-hoc* refinement, with high-quality feedback, which can produce remarkable gains in various tasks (Madaan et al., 2024; Pan et al., 2024). Text refinement offers advantages in cost efficiency (Cai et al., 2024) compared to supervised fine-tuning (SFT) for summarizer, which require substantial amounts of human-annotated training data for each domain and dimension. Moreover, this technique is further enhanced by incorporating external feedback generated by LLMs (Liu et al., 2023b; Wadhwa et al., 2024; Wan et al., 2024). Given that feedback can be easily configured as multifaceted through prompt engineering (Madaan et al., 2024), the refinement can be extended to multiple desired dimensions using external feedback.

However, extending refinement methods beyond a single dimension presents several challenges that must be addressed to be sufficient for multi-dimension. Firstly, multi-dimensional refinement must account for *trade-off* across various dimensions. What may be a correct refinement for one dimension could, in fact, be the result of sacrificing another dimension.

---

*This work is conducted independently and is not related to the author's position at Amazon.
†Corresponding author.

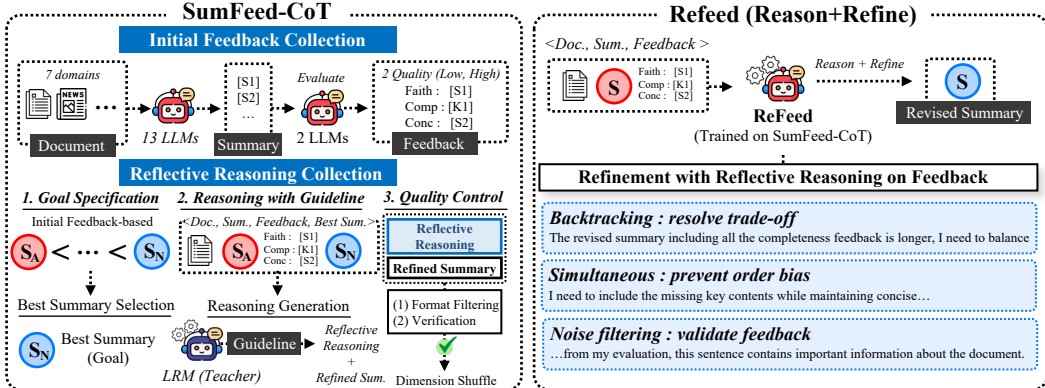

Figure 1: Overview of data construction of SumFeed-CoT, a dataset for training ReFeed and ReFeed. Faith., Comp., and Conc. denote faithfulness, completeness, and conciseness.

Secondly, there exists an *ordering bias*. Similarly to training paradigms, multi-dimensional refinement can be approached in two main ways: "sequential," where each prompt addresses one dimension at a time, or "simultaneous," a single prompt with aggregating across all dimensions. Depending on the approaches, the refinement model may exhibit a preference for the specific positioning of dimensions within the prompt (Zhang et al., 2024) or prioritize subsequent feedback. Lastly, external feedback, derived from LLM evaluations on text, may exhibit deficiencies such as low accuracy, stemming from latent biases across various domains and models (Wang et al., 2024; Ye et al., 2024). This defective feedback can be characterized as *noisy feedback*. Although refinement methods are designed to adapt to external feedback, the assumption that feedback is always accurate can be problematic. Thus, incorporating noisy feedback may inadvertently affect the target dimension and beyond.

In this paper, we address the three challenges by proposing **ReFeed** (**Re**finement with **Re**flective **Re**asoning on **Feed**back), specifically focusing on refining summaries by empha-sizing three key dimensions: faithfulness, completeness, and conciseness[1]. Our approach distills large reasoning model (LRM) such as OpenAI's o1 (Jaech et al., 2024) and integrates *reflective* reasoning on feedback into the refinement pipeline. LRMs like o1 exhibit reflective reasoning through *long chain-of-thought* (Long-CoT). Its thinking process involves "backtrack-ing," where the model identifies mistakes and restarts its reasoning (Wu et al., 2024; Zhong et al., 2024). This has the potential to revisit feedback incorporation when trade-offs emerge during refinement, simultaneously handling multiple dimensions. Furthermore, reflective reasoning enables a critical analysis of feedback, allowing for continuous validation of its correctness by tracing back through the reasoning process rather than adopting a receptive stance. Yet, given the high computational cost of such reasoning models, distilling this capability into smaller models presents a promising direction.

As shown in Figure 1, we first construct a large-scale dataset to enable complex reasoning in LLMs. Our dataset includes high-quality LLMs' reflective reasoning on feedback covering three perspectives: (i) backtracking to resolve trade-offs during the refinement; (ii) adopting a simultaneous-style refinement on multi-dimension; and (iii) validating feedback to filter out noise. We utilize reflective reasoning data from a LRM, and then curate only well-structured and successful reasoning on feedback for refinement on summaries. Lastly, we train a model on the constructed dataset (SumFeed-CoT), where input feedback undergoes shuffled permutations across multiple dimensions to better mitigate order bias. We further perform a comprehensive analysis to identify which refinement pipeline, including the prior two approaches, DCR (Wadhwa et al., 2024) and ACUEval (Wan et al., 2024), achieves a balanced improvement across multi-dimension mitigating trade-offs, and maintains robustness in the presence of order bias and noisy feedback.

---

[1]Recent works emphasize that a human-preferred summary should be factually correct (faithful-ness), cover key information (completeness), and avoid irrelevant details (conciseness) (Song et al., 2024; Tang et al., 2024b).

Our main contributions are: (1) We propose ReFeed, a multi-dimensional refinement pipeline utilizing reflective reasoning to address trade-offs, ordering bias, and noise while developing lightweight models; (2) We release SumFeed-CoT, a large-scale dataset that captures Long-CoT reasoning on feedback, enabling effective refinement by considering the three key perspectives; and (3) We design and analyze five potential refinement pipelines, including ReFeed, examining the trade-offs of prioritizing a single dimension and the vulnerabilities to order bias and noisy feedback.

## 2 Related Work

**Text Refinement.** LLM-based refinement is a widely adopted approach for enhancing text quality, leveraging either self-feedback (Madaan et al., 2024) or external feedback from other models (Paul et al., 2024; Xu et al., 2024). In summarization tasks, previous studies have primarily targeted faithfulness (Fabbri et al., 2022; Liu et al., 2023b) by designing models and pipelines to correct factual errors. More recent approaches decompose the refinement process into distinct critique and refinement modules (Zhang et al., 2023; Sun et al., 2024). However, these methods are known to introduce high error densities during refinement (Wadhwa et al., 2024), or to overly correct when applied to LLMs' summaries. In this context, a detection step is required to localize factuality errors using binary labels from automated evaluators (Wadhwa et al., 2024; Wan et al., 2024). Binary feedback has proven more effective than scalar feedback itself and, when combined with critique, offers synergistic benefits (Xu et al., 2024). Nonetheless, this advantage depends on the detector's performance, leaving the pipeline vulnerable to biases and inaccuracies in the detector model.

**Multi-Dimensional Evaluation.** Faithfulness evaluation has emerged as a more reliable approach for aligning with human preferences. Although faithfulness is a mainstream criterion for evaluating LLMs, a one-size-fits-all evaluation metric is impractical, as human preferences are inherently diverse. Thus, the multi-dimensional evaluation (Mehri & Eskenazi, 2020; Fabbri et al., 2021) is becoming the prominent regime in the evaluation for LLMs. In summarization tasks, Song et al. (2024) propose the multi-dimensional evaluation framework by introducing faithfulness, completeness, and conciseness, with clear definitions and objective criteria. Specifically, faithfulness and conciseness are evaluated at the sentence level, where faithfulness is measured by factual errors and conciseness by the inclusion of key facts in summary sentences Completeness is assessed at the key fact level, marking any missing key facts. Lee et al. (2024) benchmark LLMs on these dimensions, finding that while faithfulness remains strong, the other two dimensions often underperform in certain domains. Since previous works validate the importance of multi-dimensional evaluation, we explore extending text refinement to incorporate multiple dimensions.

## 3 Preliminary: Refinement Pipeline

### 3.1 Refinement Modules

In general, the text refinement pipeline involves three key modules, namely detection, reasoning (critique), and refinement modules (Wadhwa et al., 2024).

**Detection Module.** This employs automated evaluators like Minicheck (Tang et al., 2024a) to pinpoint parts requiring revision, such as sentences or atomic facts, using binary labels. Notably, binary labels alone can drive effective refinements (Wan et al., 2024; Wadhwa et al., 2024), making the detection label itself a form of feedback. The labels (feedback) are passed to the critique module.

**Reasoning (Critique) Module.** This generates a specific reason why correction is needed and suggests how to fix. It is often used sequentially after the detection module (Wadhwa et al., 2024) or concurrently with it (Song et al., 2024). Feedback can thus be generated by either a single module (detection or reasoning) or by critique based on detection labels.

While such step is referred to as "critique," in prior works, we instead refer to it as "reasoning" to better represent their role.

**Refinement Module.**   This refines text either without feedback or with provided feedback. Refinement is generally more effective when feedback is available rather than when performed directly without it (Madaan et al., 2024). In such cases, the feedback quality plays a crucial role in determining the effectiveness of the refinement outcome.

## 3.2   Extension to Multi-Dimension

Unlike previous refinement pipelines focusing on a single dimension, multi-dimensional feedback introduces new complexity by requiring the integration of diverse feedback at once. Here, we introduce three key aspects, A1–A3, essential for designing a multi-dimensional refinement pipeline.

**A1: Trade-off in Multi-Feedback.**   Reasoning and refinement modules should reconcile the feedback derived from multiple detection modules to address multiple dimensions, thereby making each feedback component interdependent with the others. Dependency here can act as constraints similar to multi-objective optimization (Guo et al., 2024). For instance, when addressing faithfulness and completeness at once, it is preferable to replace only erroneous phrases to preserve completeness, while deleting the entire sentence to fix a minor faithfulness issue. The latter is considered "over-correction" (Shridhar et al., 2024), leading to the loss of key information, which harm completeness. Therefore, considering trade-offs from interdependent multi-feedback is crucial in reasoning.

**A2: Sequential vs. Simultaneous.**   The refinement module can incorporate multi-dimensional feedback via two strategies inspired by training methodologies; "sequential" and "simultaneous." The sequential approach (Lou et al., 2024) provides feedback stepwise for each individual dimension, allowing concentrated attention on a target dimension that may yield higher gains. In contrast, the simultaneous approach (Wu et al., 2023; Guo et al., 2024) aggregates feedback across all dimensions into a single prompt. This ensures a more balanced consideration of all dimensions, though it lacks the concentrated attention as the sequential one.

**A3: Inaccurate Feedback.**   While reasoning and refinement modules are designed for "receptive" reasoning to feedback, it is important to note that LLM-generated feedback may be biased or inaccurate. Hence, LLMs' feedback may amplify flawed reasoning (Turpin et al., 2024). In this context, LRM like OpenAI's o1 has gained attention for its ability to perform "reflective" reasoning through Long-CoT. By iteratively reviewing feedback and retracing the reasoning, reflective reasoning can identify inaccurate feedback and mitigate the continuation of erroneous reasoning.

## 3.3   Simple Pipeline for Multi-Dimension

We design four simple pipelines addressing three key aspects, all of which serve as baselines. We integrate reasoning and refinement into a single module, referred to as "reason+refine" to foster closer alignment between the reasoning behind suggestions to fix and the refined text. In this pipeline, a document and its summary are first processed by a detection module. A reason+refine module then generates both a reasoning on feedback and a revised summary.

We introduce four baselines (P1–P4), each starting with single-dimension refinement and gradually integrating the three aspects from A1 to A3.

● **P1:** This is the base pipeline designed specifically for single-dimension feedback, following "detect→reason+refine" structure.

● **P2:** This extends P1 to support multi-dimensional feedback sequentially. The reason+refine model iterates three times, each using the improved summary from the previous run and integrating feedback for faithfulness, completeness, and conciseness in that order.

- **P3:** This is identical to P2, except for the prompt strategy. While P2 employs a single-turn prompt that resets the session at each step, P3 uses a multi-turn approach, retaining memory of previous runs for improved continuity. Compared to P2, the subsequent step in this pipeline can perceive the refinement results of the previous dimension, making the refinement results more dependent.

- **P4:** Unlike P2 and P3, this shifts sequential prompting to a simultaneous approach, providing feedback for all three dimensions.

The detailed categorization of the four baselines is presented in columns 1–5 of Table 1, and the details of the prompts are given in Appendix A.

## 4   ReFeed: Refinement with Reflective Reasoning on Feedback

ReFeed differs from the four baselines, resembling P4 (multi-dimension, simultaneous feedback) but uniquely integrating *reflective* reasoning in refinement. To distill this capability, it includes data construction and a dedicated training process. Note that, in refinement, our focus is on enhancing the quality of the initial summary across three key dimensions (Song et al., 2024; Tang et al., 2024b):

- *Faithfulness*: The factual consistency of the summary with its grounded document.

- *Completeness*: The coverage of key information in the summary.

- *Conciseness*: The succinctness of the summary, avoiding unnecessary details.

We present the details of them in Appendix B.

### 4.1   SumFeed-CoT Dataset

As existing LRMs are mainly trained on structured problems, such as solving math problems and code generation, they exhibit some misalignment with the need for summarization refinement. Thus, we guide the teacher model to refine summary based on reflective reasoning on feedback, thereby constructing a training dataset. Motivated by previous works (Jaech et al., 2024; Xu et al., 2025a), we construct our dataset through three stages in Figure 1: *Goal Specification*, *Guideline Formulation*, and *Quality Control*.

A well-defined goal, such as a golden label, provides a clear reference for the LRM, steering it toward an optimal reasoning path. Detailed guidelines further reinforce this process by ensuring alignment between the reasoning trajectory and the intended objective (Jaech et al., 2024). Finally, rigorous quality control is essential for curating high-quality reasoning data (Lu et al., 2024), which, in turn, enhances student models' ability to engage in reflective reasoning for refinement.

We outline the three stages for data construction in the following sections.

#### 4.1.1   Initial Feedback Collection

We generate initial feedback for reasoning using document-summary pairs. Following Song et al. (2025), we sample 200 documents from the train split of seven datasets across diverse domains and generate summaries using 13 different models to ensure diversity of summary conditions. Each summary is evaluated with binary labels that identify sentences to fix for faithfulness or conciseness, or missing key facts that should be added for completeness through LLM-based evaluator[2]. To address various quality feedback, we adapt the backbone of the LLM-based evaluator using both LLaMA-3.1-8B (low quality) and -70B (high quality) models. We then map the binary labels from the automatic evaluator to corresponding sentences and key facts, each of which serves as feedback to the refinement model.

After filtering feedback's format errors, we obtain 29K <document, summary, feedback> triplets, each serving as input to the reason+refine model for summary refinement. Refer to Appendix C for details on source data, summarizers, and prompts.

---

[2]We used the most recent automated evaluator tailored for summary, FineSurE (Song et al., 2024).

### 4.1.2 Reflective Reasoning Collection

Summary refinement currently lacks a clear goal and guidelines. To address this, we define the ideal summary as one that needs no further feedback and concisely captures all key facts. We also establish guidelines to address three key aspects of multi-dimensional refinement: (1) backtracking for trade-offs, (2) simultaneous style to avoid order bias and (3) noise filtering. Lastly, we curate high-quality reasoning via two criteria: *format filtering*, which retains Long-CoT without formatting errors, and *verification-based filtering*, which ensures that the reasoning meets established quality standards. The reasoning collection process in our dataset is detailed as follows:

**Goal Specification.**   We define the best summary as the one selected from those generated by the 13 summarizers during the feedback collection step. For each document, the best one is the summary with high quality that only needs minimal feedback. Based on the initial feedback, we select the summary with the highest average across faithfulness, completeness, and conciseness as the best one.

**Reasoning with Guideline.**   We define a guideline to distill reflective reasoning from a tearcher LRM[3] for summary refinement. The reasoning is collected by providing <document, summary, feedback, best summary> as input to the teacher LRM, along with our well-formed guideline, to induce reflective reasoning. The reasoning derives how to refine the given summary by referencing the best summary (goal) as a standard.

Specifically, the guideline prompts the teacher LRM to reason across multiple dimensions simultaneously, incorporating reflective reasoning such as backtracking to manage trade-offs and validating feedback. Additionally, it encourages the model to analyze the best summary's attributes, allowing it to better filter out noisy feedback in the input on completeness and conciseness. Yet, this does not enhance the faithfulness critique. Thus, we introduce explicit factual error definitions, aiding more accurate faithfulness feedback assessment (Wadhwa et al., 2024). See Appendix D for details.

**Quality Control.**   We perform format filtering to retain only samples with over 5K tokens in reasoning after removing malformed outputs , ensuring format compliance and compatibility with the student model's token limit. Next, we conduct a verification-based filtering, verifying the revised summary with the collected reasoning indeed meets the following criteria: (1) faithfulness = 1, completeness $\geq$ 0.5, and conciseness $\geq$ 0.5; and (2) $\triangle$ score is $> 0$ across all dimensions, meaning that the revised summary must improve upon the original one in every dimension.

Consequently, our final dataset includes 7.7K training samples, each with <document, summary, feedback> for the input prompt and <reflective reasoning, refined summary> for the output response. We then randomly shuffle the order of dimensions in the feedback to mitigate order bias. The details of training data can be found in Appendix E.

### 4.2 Training and Inference Details

We fine-tune LLaMA-3.1-8B-Instruct (student) using LoRA (Hu et al., 2022) and DeepSpeed (ZeRO-3) (Rasley et al., 2020). The user prompt follows the same format as reasoning generation, excluding the best summary (i.e., goal) and error types. At inference time, the model generates both reasoning and a refined summary enclosed within separate XML tags, as shown below. For detailed input/output formats, training parameters, and inference prompt, refer to Appendix F.

---

**Summary of Input/Output in ReFeed**

Input: Document, Summary, Feedback
Output: <think> Reflective Reasoning </think> <answer> Refined Summary </answer>

---

[3]We used QwQ-32B-preview model, which exhibits Long-CoT capabilities comparable to o1-mini.

| Pipeline | Dim. | Dep. | Simul. | Reflect. | Faith. | Comp. | Conc. | Avg. |
|---|---|---|---|---|---|---|---|---|
| Before Refine | | | | | 78.0 | 46.4 | 76.4 | 66.9 |
| P1-Faith | Faith. | ✗ No | ✗ No | ✗ No | 80.7*(+2.7) | 45.9 (-0.5) | 80.4*(+4.0) | 69.0*(+2.1) |
| P1-Comp | Comp. | ✗ No | ✗ No | ✗ No | 80.4*(+2.4) | **62.0***(+15.6) | 79.0*(+2.6) | 73.8*(+6.9) |
| P1-Cons | Conc. | ✗ No | ✗ No | ✗ No | 76.7 (-1.3) | 44.2 (-2.2) | **86.3***(+9.9) | 69.1*(+2.2) |
| P2 | Multi | ✗ No | ✗ No | ✗ No | 78.4*(+0.4) | 51.5*(+5.1) | 84.8*(+8.4) | 71.6*(+4.7) |
| P3 | Multi | ✓ Yes | ✗ No | ✗ No | 78.9 (+0.9) | 53.2*(+6.8) | 80.0*(+3.6) | 70.7*(+3.8) |
| P4 | Multi | ✓ Yes | ✓ Yes | ✗ No | 80.1*(+2.1) | 56.0*(+9.6) | 83.6*(+7.2) | 73.2*(+6.3) |
| ReFeed | Multi | ✓ Yes | ✓ Yes | ✓ Yes | **82.7***(+4.7) | 60.0*(+13.6) | 83.4*(+7.0) | **75.3***(+8.4) |

Table 1: Summary quality of revised summaries on three dimensions across seven refinement pipelines. "Dep.", "Simul.", and "Reflect." denote feedback interdependency, simultaneous approach, and reflective reasoning, respectively. In the sequential approach (P2, P3), feedback is incorporated in the order of faithfulness→completeness→conciseness. The best scores are marked in bold. * denotes a statistically significant difference ($p < 0.05$) based on a paired bootstrap test.

# 5 Evaluation

## 5.1 Experiment Setup

**Test Dataset.** We compare ReFeed with four baselines (P1–P4) on UniSumEval (Lee et al., 2024), which features challenging source documents in nine domains, with summaries from nine summarizers (both non-LLMs and LLMs) prone to factual errors, missing information, and unnecessary content.

Additionally, since refinement tasks require feedback labels aligned with the input summary, we collect initial feedback labels for UniSumEval's summaries using FineSurE (Song et al., 2024) (as a "detector" module). This generates feedback, including sentence-level labels for faithfulness and conciseness and key fact-level labels for completeness for each <document,summary> pair. Note that two types of feedback are used to adjust the quality in FineSurE by employing different backbones: LLaMA-3.1-8B/70B for *low-* and *high-*quality feedback, respectively. ReFeed and all baselines receive the same input for refinement: <document,summary,feedback>. The goal is to refine the initial summary using LLM-generated feedback and produce a revised one.

**Evaluation Metric.** We assess the quality of the refined summaries using FineSurE again, this time with GPT-4o as the backbone. GPT-4o is chosen for its high accuracy in evaluation, ensuring reliable assessments, unlike input feedback, which may contain noise. We assess the revised summary on faithfulness, completeness, and conciseness. We also provide results with humans and another automated evaluator, G-Eval, in Appendices H and G, along with the metric details.

**Compared Methods.** Since existing methods focus on single-aspect refinement, we primarily compare the multi-dimensional pipelines, P1–P4, in Section 3.3. For a complete evaluation, we also evaluate two recent methods, DCR (Wadhwa et al., 2024) and ACUEval (Wan et al., 2024) in Section 5.6, though both are designed only for a single dimension, i.e., faithfulness.

**Main Questions.** Our experiment is mainly designed to address the following three research questions.

**RQ1.** We compare refinement pipelines based on their capacity for dimensions (single vs. multi), feedback exposure (sequential vs. simultaneous), and reasoning policy (receptive vs. reflective).

| Pipeline | Random | | | Last–Faith | | | Last–Comp | | | Last–Conc | | | Max–Min | | |
|---|---|---|---|---|---|---|---|---|---|---|---|---|---|---|---|
| | FA | CM | CN | FA | CM | CN | FA | CM | CN | FA | CM | CN | FA | CM | CN |
| P2 | 80.8 | 59.1 | **85.3** | 80.9 | 59.0 | **85.0** | 81.1 | **63.2** | 83.9 | 79.6 | 55.3 | **85.1** | 1.5 | 7.9 | 1.4 |
| P3 | 80.7 | 59.3 | 82.8 | 80.8 | 57.7 | 82.5 | 80.4 | 62.5 | 82.4 | 79.5 | 58.4 | 83.0 | 1.3 | 4.8 | 0.6 |
| P4 | 83.8 | 61.3 | 85.1 | 82.9 | 60.6 | 84.9 | **83.6** | 60.7 | 83.8 | 81.9 | 59.8 | 84.8 | 1.9 | 1.5 | 1.3 |
| ReFeed | **84.2** | **62.9** | 84.2 | **83.6** | **62.6** | 84.6 | 83.4 | 62.4 | **84.5** | **84.3** | **62.9** | 84.4 | **0.9** | **0.5** | **0.4** |
| ReFeed(–) | **84.2** | 61.1 | 85.2 | 83.3 | 60.1 | 84.2 | 82.7 | 60.1 | **84.5** | 83.1 | 61.6 | 84.5 | 1.5 | 1.5 | 0.7 |

Table 2: Summary quality after refinement with multi-dimensional refinement pipelines in four shuffling setups. FA, CM, and CN denote faithfulness, completeness, and conciseness. "Max–Min" is the score gap between the highest and lowest values for each dimension, with smaller ones indicating better robustness (the smallest value in bold).

**RQ2.** We examine how the order of feedback dimensions affects the refinement quality, evaluating how robustly the refinement process incorporates multi-dimensional feedback, regardless of order.

**RQ3.** We examine how noisy feedback affects the quality of refined summaries. This highlights how detection errors can cascade into refinement outcomes, emphasizing the importance of reflective reasoning over receptive one.

The following subsections provide an in-depth analysis of the three main research questions.

### 5.2 RQ1: Refinement Pipeline Comparison

Table 1 presents the revised summary quality across five refinement pipelines, namely P1 to P4 and ReFeed (Ours), along with the quality of the original summaries in UniSumEval before refinement. All refinement pipelines use the LLaMA-3.1-8B backbone and take low-quality feedback, along with the document and summary, as input.

**Highlight.** ReFeed outperforms other pipelines in the average score ("Avg."), a composite measure across three dimensions. This indicates that **ReFeed effectively balances trade-offs, accounts for feedback dependencies, and remains robust to detector inaccuracies.** Also, this trend is consistent in human evaluation; refer to Appendix H.

#### 5.2.1 Detailed Analysis

*Feedback Dependency:* While P1-Faith and P1-Conc achieve significant improvements in faithfulness and conciseness, respectively, they reduce completeness or faithfulness. This confirms that **focusing solely on one dimension compromises others** due to the dependencies on multi-feedback.

*Single- vs. Multi-turn:* In the sequential refinement scheme, **the single-turn method (P2) exhibits a large disparity in quality improvements across dimensions**, as it resets previous sessions during refinement. Specifically, P2 maximizes conciseness (the last dimension addressed) over the other two dimensions. In contrast, the multi-turn approach (P3) achieves more balanced improvements, though the gains remain modest, i.e., a similar composite score for P2 and P3.

*Sequential vs. Simultaneous:* Employing multi-dimension feedback sequentially or simultaneously (P3–P4) leads to reasonable improvements across all three dimensions. But, the simultaneous approach (P4) outperforms sequential strategies (P3) on their composite scores. Therefore, **refining in a single prompt is more effective than splitting the process across multiple turns**, as it prevents information loss with each turn change.

*Receptive vs. Reflective:* ReFeed (reflective reasoning) shows a notable improvement over P4 (receptive reasoning), increasing faithfulness by 2.6 and completeness by 4.0. This large gain confirms that **reflective reasoning is essential for summary refinement to mitigate multi-faceted risks and manage feedback interdependencies.**

### 5.3 RQ2: Robustness against Ordering Bias

Another key aspect of a robust refinement pipeline is resilience to feedback order variations, ensuring consistent refinement quality. Thus, unlike the previous setup in Section 5.2, we design four shuffling variants, each following a different policy: *Random*, where feedback order is fully shuffled across three dimensions at the sample level, ensuring each input has a random order; and *Last-Dim*, where one dimension is fixed in the final position while the first two are randomly shuffled, resulting in three setups: *Last-{Faith,-Comp,Cons}*.

Table 2 shows the summary quality's variation over the four setups. In this experiment, we use high-quality feedback as input for summary refinement to minimize bias from noisy feedback. Note that ReFeed(–) is a variant of ReFeed fine-tuned on non-shuffled SumFeed-CoT data to assess the impact of feedback dimension shuffling in training.

ReFeed demonstrates greater stability due to its refinement strategy and training method. In sequential pipelines (P2 and P3), completeness scores fluctuate significantly, reflected in the high Max–Min values of 7.9 and 4.8, respectively. **This variability likely stems from order bias, i.e., later dimensions overly influence refinement performance**, as evidenced by P2's superior scores on its final dimension. Additionally, the independent reflection strategy in P2 may amplify this fluctuation.

Lastly, **shuffling the dimension order in the training data mitigates ordering bias**, leading to improved overall performance and reduced variance, as shown by the performance gap between ReFeed and ReFeed(–).

### 5.4 RQ3: Robustness against Feedback Quality

An ideal refinement pipeline should improve the summary to high quality through reasoning, regardless of the feedback quality. To validate this, we vary the feedback between low and high quality, enabling us to evaluate performance consistency, as summarized in Table 3.

ReFeed achieves the most consistent scores, regardless of the feedback quality. Specifically, ReFeed shows a score drop of 0.06–0.25 across all three dimensions, while even the second-best method (P4) shows a drop of 0.12–4.0. In particular, ReFeed, even with low-quality feedback, outperforms P1-Faith, which focuses solely on faithfulness, despite P1 receiving high-quality feedback. Even worse, the P1 series struggles even in their targeted dimensions, making them

| Pipeline | Faith. | | Comp. | | Conc. | |
|---|---|---|---|---|---|---|
| | High | Low | High | Low | High | Low |
| Before. | 78.0 | | 46.4 | | 76.4 | |
| P1-Faith | 82.2* | 80.7* | 47.1* | 45.9 | 79.7* | 80.4* |
| P1-Comp | 82.7* | 80.4* | **66.2*** | **62.0*** | 80.9* | 79.0* |
| P1-Conc | 78.9 | 76.7 | 45.5* | 44.2* | **86.9*** | **86.3*** |
| P4 | 83.3* | 80.1* | 60.0* | 56.0* | 84.8* | 83.6* |
| ReFeed | **84.6*** | **82.7*** | 62.5* | 60.0* | 84.0* | 83.4* |

Table 3: Summary quality of revised summaries under low- and high-quality feedback. "Before." indicates scores before refinement. * denotes a statistically significant difference ($p < 0.05$) based on a paired bootstrap test.

vulnerable to feedback quality. That is, the receptive reasoning in P1 and P4 does not ensure robustness to feedback quality. Therefore, **ReFeed's reflective reasoning achieves high robustness to feedback quality and produces high-quality summaries.**

### 5.5 Further Study on Reflective Reasoning

We conduct an in-depth investigation to demonstrate the effectiveness of reflective reasoning over receptive one. To do so, we compare two student models using the LLaMA-3.1-8B backbone distilled from the same teacher (QwQ-32B-preview) but trained with different reasoning strategies, i.e., $P4_{FT}$ (receptive), which is a variant of P4 fine-tuned, and ReFeed (reflective). To fine-tune P4, we create a dataset following the same process as SumFeed-CoT with P4's prompt for receptive reasoning. Refer to Appendix I for details of training data and prompt for P4.

| Model | Pipeline | Composite | |
|---|---|---|---|
| | | High | Low |
| LLaMA-3.1 8B | P4 | 57.0* | 54.9* |
| | P4$_{4k}$ | 55.3* | 54.3* |
| | ReFeed$_{4k}$ | 57.1* | 55.7* |
| | ReFeed$_{8k}$ | **57.8*** | **56.5*** |
| QwQ 32B (Teacher) | | 58.1* | 57.3* |

Table 4: Comparison of reflective reasoning over receptive one on low- and high-quality feedback setups. P4$_{4k}$ denotes P4 fine-tuned on 4K data; 4K/8K: training dataset size.

| Pipeline | Faith. | Comp. | Conc. |
|---|---|---|---|
| Before | 78.0 | 46.4 | 76.4 |
| ACUEval | 80.8* (+2.8) | 43.8* (-2.6) | 81.9* (+5.5) |
| DCR | 81.2* (+3.2) | 42.8* (-3.6) | 76.7 (+0.3) |
| ReFeed | **82.7*** (+4.7) | **60.0*** (+13.6) | **83.4*** (+7.0) |

Table 5: Comparison with the previous pipelines, which focused only on faithfulness. * indicates a statistically significant difference (p < 0.05, paired bootstrap test).

Table 4 presents the improvement of P4 after fine-tuning compared to ReFeed. Since P4's final dataset size is 5K, we train two models, P4$_{FT}$ and ReFeed, using only the 4K overlapping data, where the latter is denoted as ReFeed$_{4K}$.

It is of interest to see that **refinement performance worsens after fine-tuning P4**, likely due to the absence of explicit goals and critique guidelines, such as best summaries and factual error types in ReFeed's training process. Moreover, we observe that ReFeed improves as the increase of the data size from 4K to 8K. These results highlight the importance of both carefully crafted reasoning guidelines and large data volumes for boosting reasoning capabilities, consistent with prior works (Jaech et al., 2024; Xu et al., 2025b).

From a distillation perspective, we compare the refinement performance and latency of ReFeed with those of the teacher, which is prompted with ReFeed's prompt without training. ReFeed achieves nearly equivalent refinement performance to the teacher, while achieving inference speeds approximately 4x faster (see latency comparison in Appendix I.3).

Furthermore, in Appendix I.5, we compares a refinement model trained on the revised summaries in SumFeed-CoT, which directly refines summaries without using the detection and reasoning modules. While this direct refinement model optimizes heavily for one dimension and shows only minor gains in others, ReFeed achieves more balanced improvements across all aspects. This underscores that reflective reasoning on multiple feedback contribute to balanced refinement quality beyond what training on revised summaries can achieve.

### 5.6 Comparison with Existing Pipeline

For a complete evaluation, we compare ReFeed with two recent summary refinement methods, DCR (Wadhwa et al., 2024) and ACUEval (Wan et al., 2024), despite their original focus solely on enhancing faithfulness. To ensure a reasonably fair comparison, we use the same LLaMA-3.1-8B backbone for refinement and provide input feedback generated by the same model used in ReFeed.

ReFeed demonstrates improvements across all dimensions without sacrificing performance in any dimensions via reflective reasoning. On the other hand, **DCR and ACUEval incur an alignment tax in completeness**, despite improving faithfulness. This faithfulness–completeness trade-off aligns with the findings in Song et al. (2025). These results highlight the necessity of balancing trade-offs for multi-dimensional feedback refinement.

## 6 Conclusion

We propose ReFeed for enhancing summary refinement across multiple dimensions. Our experiments show that incorporating reflective reasoning into the refinement process outperforms four alternatives and existing approaches, by achieving a better balance across dimensions while keeping robustness against order bias and noise in feedback. Also, we observe that ReFeed delivers refinement performance comparable to its teacher model while significantly reducing inference time.

## Ethics Statement

Our work primarily focuses on reasoning capabilities based on diverse LLM-generated feedback, which does not raise any ethical concerns during the model training phase.

## Acknowledgements

This research was supported by KISTI in 2025 (No.(KISTI) K25L1M1C1), aimed at developing KONI (KISTI Open Neural Intelligence, a large language model specialized in science and technology), and by the NRF funded by MSIT (No. RS-2022-NR068758). For GPU infrastructure, our work was supported by the IITP grant funded by MSIT (No. RS-2025-02653113, High-Performance Research AI Computing Infrastructure Support at the 2 PFLOPS Scale).

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

| Pipeline | Dim. | Feedback | Refinement | Reasoning |
|----------|------|----------|------------|-----------|
| P1 | Single | Independent | Sequential | Receptive |
| P2 | Multi | Independent | Sequential | Receptive |
| P3 | Multi | Dependent | Sequential | Receptive |
| P4 | Multi | Dependent | Simultaneous | Receptive |
| ReFeed | Multi | Dependent | Simultaneous | Reflective |

Table 6: Five different pipelines for refinement on multi-dimensions.

## A  Summary of Refinement Pipelines

### A.1  Refinement Prompts

Table 6 presents the configurations of five distinct pipelines. Each reason+refine module in these pipelines receives three inputs: a document, summary, and feedback. The instruction to reason feedback across three dimensions is provided in prompt, and the order of feedback and instruction is interchangeable. Tables 17–19 present the prompts for each pipeline. We also provide the previous two refinement pipelines DCR (Wadhwa et al., 2024) and ACUEval (Wan et al., 2024) in Tables 20–21, ensuring that their refinement prompts follow the same format as ours, while the remaining prompts are directly adopted from the original papers.

## B  Metric for Summary Quality

By recent study (Song et al., 2024), we employ three core metrics—faithfulness, completeness, and conciseness to evaluate summary quality.

**Faithfulness Score.**   The faithfulness metric is determined by examining sentence-level factual correctness. Let $S = \{s_1, \ldots, s_N\}$ be a summary composed of $N$ sentences, where each $s_i$ is the $i$-th sentence. Define $S_{\text{fact}} \subseteq S$ as the subset of sentences verified as "factually correct." The faithfulness score of $S$ with respect to the source document $D$ is given by:

$$\text{Faithfulness}(D, S) = \frac{|S_{\text{fact}}|}{|S|}. \tag{1}$$

This measures the fraction of sentences in the summary that are deemed factually accurate relative to the total number of sentences.

**Completeness and Conciseness Score.**   Let $K = \{k_1, \ldots, k_M\}$ represent a set of key facts, where $M$ denotes the total count of these facts. By leveraging alignments between key facts and summary sentences, we construct a bipartite graph $M = (K, S, E)$, where the edge set $E = \{(k, s) : k \rightarrow s \mid k \in K \wedge s \in S\}$. Here, $k \rightarrow s$ indicates that the key fact $k$ appears within sentence $s$ in the summary. Based on this graph, the completeness and conciseness scores for the summary $S$ are computed as:

$$\text{Completeness}(K, S) = \frac{|\{k \mid (k, s) \in E\}|}{|K|}, \tag{2}$$

$$\text{Conciseness}(K, S) = \frac{|\{s \mid (k, s) \in E\}|}{|S|}. \tag{3}$$

Here, $|\cdot|$ denotes set cardinality. The completeness score captures the extent to which key facts are included in the summary, whereas the conciseness score evaluates how effectively the summary condenses and integrates those key facts.

**Composite Score.**   For scenarios involving multi-dimensional feedback, we calculate the average of the three percentage scores—faithfulness, completeness, and conciseness—to derive a composite score, which is used to assess a summary's comprehensive quality.

| Model Name | Checkpoints |
|---|---|
| Bart-large | facebook/bart-large-cnn |
| Pegasus-large | google/pegasuscnn_dailymail |
| Flan-t5-large | spacemanidol/flan-t5-large-cnndm |
| Phi-2 | microsoft/phi-2 |
| Mistral-7b-inst | mistralai/Mistral-7B-Instruct-v0.2 |
| Mixtral-8x7b-inst | mistralai/Mixtral-8x7B-Instruct-v0.1 |
| Llama2-13b-chat | meta-llama/Llama-2-13b-chat-hf |
| Mistral-nemo | mistralai/Mistral-Nemo-Instruct-2407 |
| Gemma2-27b-inst | google/gemma-2-27b-it |
| Llama3-70b | meta-llama/Meta-Llama-3-70B-Instruct |
| Claude-instant | claude-instant (AWS Bedrock) |
| GPT-3.5$_{turbo}$ | gpt-3.5-turbo-0125 (OpenAI) |
| GPT-4$_{turbo}$ | gpt-4-0125-preview (OpenAI) |

Table 7: Checkpoints of the 13 summarizers. For open-source models, we use publicly available checkpoints from Huggingface, while for proprietary models, we utilize paid API services by OpenAI and AWS Bedrock.

---

**Text**: {input text}

**Instruction**: Summarize the Text.

Provide your answer in JSON format. The answer should be a dictionary with the key "summary" containing a generated summary as a string: {"summary": "your summary"}

**JSON Output**:

---

Table 8: Prompt to generate summary

## C   Initial Feedback Creation Details

### C.1   Input Document Sourcing

Acquiring feedback from a diverse range of source documents is essential for thoroughly evaluating the limitations of contemporary text summarizers, particularly regarding variations in input domain, text length, and format (dialogue vs. non-dialogue) (Lee et al., 2024; Song et al., 2025). Following Song et al. (2025), we compile input texts from multiple datasets that span 7 distinct domains, covering a spectrum from brief to extensive texts and including both non-dialogue and dialogue formats. Specifically, we randomly sample 200 texts from the training set of each dataset, which comprises four non-dialogue sources—CNN/DM (news) (Nallapati et al., 2016), Wikihow (lifestyle) (Koupaee & Wang, 2018), GovReport (report) (Huang et al., 2021), and PubMed (medical literature) (Cohan et al., 2018)—as well as three dialogue datasets—DialogSum (dailylife) (Chen et al., 2021), MediaSum (interview) (Zhu et al., 2021), and MeetingBank (meeting) (Hu et al., 2023).

### C.2   Summary Generation

We produce summaries of varying quality and distribution by leveraging a suite of 13 different language models, which are listed in Table 7. To ensure a consistent and reproducible approach to summary generation, we employ specific prompts that are detailed in Table 8.

### C.3   Initial Feedback Generation

We employ FineSurE (Song et al., 2024) to obtain multi-dimensional and fine-grained initial feedback. Specifically, feedback on faithfulness is collected at the sentence level, while completeness and conciseness are assessed at the key fact level. FineSurE utilizes LLMs

to conduct a fact-checking task for faithfulness and a key fact alignment task for the other two dimensions. Since source datasets do not include human-annotated key facts, we also extract the list of key facts in document using the prompt introduced in Song et al. (2024). The prompts used for FineSurE are provided in Table 22–23.

The binary labels generated by this step are utilized in the reason+refine module. Additionally, these labels are aggregated into three percentage scores—each corresponding to one of the evaluated dimensions—to assess summary quality, as described in Appendix B.

## D    Reasoning Creation Details

We map the initial feedback, provided as binary labels, where a label of 1 indicates that refinement is needed, into a simple textual structure corresponding to the three dimensions, as detailed below:

---

**Mapping of feedback**

**Summary Sentences:** [S1, S2, S3]
**Key Facts:** [K1, K2, K2]

**Faithfulness Feedback:** [1, 0, 0]
**Completeness Feedback:** [1, 1, 0]
**Conciseness Feedback:** [0, 0, 1]

**Mapping:**

Faithfulness Feedback:
These summary sentences are factually inconsistent with the Document:
- [S1]

Completeness Feedback:
These key contents are missing in the summary:
- [K1]
- [K2]

Conciseness Feedback:
These summary sentences do not contain key content:
- [S3]

---

Table 24 shows the prompt used to generate Long-CoT data in SumFeed-CoT. In this prompt, the inputs consist of the document, summary, and detection labels transformed into verbal form. The output is generated through reflective reasoning and ends with a revised summary presented in a boxed format.

## E    Dataset Details

We present a detailed statistical analysis of the SumFeed-CoT dataset, which consists of <document, summary, feedback> triplets as input and <reasoning + revised summary> as output. Each reasoning in the dataset is a Long-CoT with more than 5K tokens. Table 9 presents statistics of the SumFeed-CoT training set, including the average token counts for documents, summaries, and reasoning steps, as well as the distribution of feedback labels across faithfulness, completeness, and conciseness dimensions.

| # of Documents | Document Token | Summary Token | Reasoning Token | Faith. Feedback | Comp. Feedback | Conc. Feedback |
|---|---|---|---|---|---|---|
| 7713 | 708.2 (14–4972) | 76.6 (2–460) | 1393.5 (523–4470) | 3.2 (1–17) | 2.6 (0–17) | 1.4 (0–11) |

Table 9: Statistics of the SumFeed-CoT training set, showing the average token counts for documents, summaries, and reasoning based on the LLaMA-3.1 tokenizer, along with the average number of feedback labels across three dimensions: faithfulness, completeness, and conciseness. The corresponding min-max ranges are provided in parentheses below each value.

| Parameter | Value |
|---|---|
| Backbone model | LLaMA-3.1-8B-Instruct |
| Batch size | 32 |
| Epochs | 3 |
| Learning rate | 1e-4 |
| Max sequence length | 10,000 |
| LoRA rank/alpha | 16 / 32 |
| Warmup ratio | 0.03 |
| LR scheduler type | cosine |

Table 10: Hyperparameters of the training configuration.

## F   Training and Inference Detail

We fine-tune LLaMA-3.1-8B-Instruct using LoRA (Hu et al., 2022) and DeepSpeed (ZeRO-3) (Rasley et al., 2020) on 4 NVIDIA L40S GPUs. Table 10 summarizes the training configurations for SFT. We incorporate a system prompt to promote long-CoT-style generation following Min et al. (2024). We use the same prompt for user prompt excluding the best summary and error types to train student model. In the training phase, the revised summary is wrapped with the <answer> tag, while all preceding intermediate reasoning steps are encapsulated in <think> tags. Both the best summary and the error type are omitted from the training process, as such details may not be available during the inference phase. Table 25 shows an example of our prompt for training.

Table 26 presents the inference prompt for ReFeed. In the inference phase, the model trained on SumFeed-CoT generates a revised summary and reasoning in the same format as in the training phase. The model's reasoning process involves backtracking and filtering out noisy feedback, enabling it to effectively process multi-dimensional feedback while refining its summary towards what it perceives as the best summary.

## G   Automated Evaluation Details

To evaluate the quality of revised summaries, we use the same FineSurE prompt described in Section C.3. Additionally, we perform automated evaluations using G-Eval (Liu et al., 2023a), which assigns a scalar score from 1 to 5 to assess summary quality. We use a customized G-Eval prompt with LLaMA-3.1-70B-instruct to assess faithfulness, completeness, and conciseness, following Song et al. (2024)

### G.1   Automated Evaluation using G-Eval

Table 11 presents the automated evaluation results using G-Eval. Despite the change in evaluation metric, the trade-offs introduced by single-dimension refinement and the bias induced by order remain consistent. Notably, ReFeed demonstrates superior performance compared

| Pipeline | Dim. | Dep. | Simul. | Reflect. | Faith. | Comp. | Conc. | Avg. |
|---|---|---|---|---|---|---|---|---|
| | Before Refine | | | | 4.22 | 3.04 | 3.42 | 3.55 |
| P1-Faith | Faith. | ✗ No | ✗ No | ✗ No | 4.33*(+0.11) | 3.03 (-0.01) | 3.46*(+0.04) | 3.60*(+0.05) |
| P1-Comp | Comp. | ✗ No | ✗ No | ✗ No | 4.44*(+0.22) | **3.69***(+0.65) | 3.59*(+0.17) | 3.90*(+0.35) |
| P1-Cons | Conc. | ✗ No | ✗ No | ✗ No | 4.42 (+0.20) | 2.97*(-0.07) | **3.66***(+0.24) | 3.68*(+0.13) |
| P2 | Multi | ✗ No | ✗ No | ✗ No | 4.46*(+0.24) | 3.28*(+0.24) | 3.67*(+0.23) | 3.80*(+0.25) |
| P3 | Multi | ✓ Yes | ✗ No | ✗ No | 4.39 (+0.17) | 3.32*(+0.28) | 3.58*(+0.16) | 3.76*(+0.21) |
| P4 | Multi | ✓ Yes | ✓ Yes | ✗ No | 4.44*(+0.22) | 3.41*(+0.37) | 3.64*(+0.22) | 3.83*(+0.28) |
| ReFeed | Multi | ✓ Yes | ✓ Yes | ✓ Yes | **4.53***(+0.29) | 3.62*(+0.58) | 3.68*(+0.26) | **3.94***(+0.39) |

Table 11: Summary quality of revised summaries on faithfulness, completeness, and conciseness across seven refinement pipelines using G-Eval. The best scores are marked in bold. * denotes statistically significant difference with $p < 0.05$ according to a paired bootstrap test.

| Pipeline | Dim. | Dep. | Simul. | Reflect. | Faith. | Comp. | Conc. | Avg. |
|---|---|---|---|---|---|---|---|---|
| | Before Refine | | | | 75.2 | 43.4 | 74.5 | 64.4 |
| P1-Faith | Faith. | ✗ No | ✗ No | ✗ No | **79.3***(+4.1) | 42.6* (-0.8) | 77.1*(+2.6) | 66.3*(+1.9) |
| P1-Comp | Comp. | ✗ No | ✗ No | ✗ No | 74.8 (-0.4) | **57.7***(+14.3) | 76.0*(+1.5) | 69.5*(+4.1) |
| P1-Cons | Conc. | ✗ No | ✗ No | ✗ No | 77.1*(+1.9) | 40.7*(-2.7) | **83.0***(+8.5) | 66.9*(+2.5) |
| P2 | Multi | ✗ No | ✗ No | ✗ No | 72.7*(-2.5) | 48.3*(+4.9) | 81.3*(+6.8) | 67.4*(+3.0) |
| P3 | Multi | ✓ Yes | ✗ No | ✗ No | 73.7* (-1.5) | 50.0*(+6.6) | 77.4*(+2.9) | 67.0*(+2.6) |
| P4 | Multi | ✓ Yes | ✓ Yes | ✗ No | 76.5 (+1.3) | 52.1*(+8.7) | 80.4*(+5.9) | 69.7*(+5.3) |
| ReFeed | Multi | ✓ Yes | ✓ Yes | ✓ Yes | 77.2*(+2.0) | 55.6*(+12.2) | 80.5*(+6.0) | **71.1***(+6.7) |

Table 12: Summary quality of revised summaries on faithfulness, completeness, and conciseness across seven refinement pipelines using LLaMA-3.1-70B-instruct as the FineSurE's automated evaluator. The best scores are marked in bold. * denotes statistically significant difference with $p < 0.05$ according to a paired bootstrap test.

to other pipelines, observed in Section 5. This demonstrates proficient performance and robustness of ReFeed across different evaluation frameworks.

## G.2 Automated Evaluation using Different LLM

We also use another LLM, LLaMA-3.1-70B-instruct, for FineSurE, to confirm the performance of each pipeline. Table 12 presents the automated evaluation results for the revised summary obtained by replacing the backbone model for FineSurE with LLaMA-3.1-70B-instruct. Although the scores across three dimensions were somewhat lower, the overall trends were consistent with those observed using other evaluation metrics. Moreover, the evaluation reveals a more pronounced order bias in P2 and P3 as indicated by the lower faithfulness scores.

## H Results by Human Evaluation

We conduct a human evaluation in Section 5.2 to determine which multi-dimensional refinement pipeline best aligns with human judgment, excluding pipeline P1, which is a pipeline for a single dimension. For human evaluation, we randomly sample 15% of the documents from the test dataset and assess the revised summary based on the model summary for these documents.

| Pipeline | Dim. | Dep. | Simul. | Reflect. | Faith. | Comp. | Conc. | Avg. |
|---|---|---|---|---|---|---|---|---|
| | | | Before Refine | | 77.4 | 42.8 | 80.6 | 66.9 |
| P2 | Multi | ✗ No | ✗ No | ✗ No | 77.3 (-0.1) | 45.8 (+3.0) | 84.5 (+3.9) | 69.2 (+2.3) |
| P3 | Multi | ✓ Yes | ✗ No | ✗ No | 77.8 (+0.4) | 50.8 (+2.8) | 79.3 (+2.4) | 69.3 (+2.4) |
| P4 | Multi | ✓ Yes | ✓ Yes | ✗ No | 84.5 (+7.1) | 52.0 (+9.2) | 81.1 (+0.5) | 72.5 (+5.3) |
| ReFeed | Multi | ✓ Yes | ✓ Yes | ✓ Yes | **87.3** (+9.9) | **54.2** (+11.4) | **88.7** (+8.1) | **76.7** (+9.8) |

Table 13: (Human evaluation) summary quality of revised summaries on faithfulness, completeness, and conciseness across seven refinement pipelines based on human evaluation. The best scores are marked in bold.

**Annotation Protocol.** Our human evaluation follows the protocol in (Lee et al., 2024). Specifically, human annotators assess faithfulness through a fact verification task, in which they identify the presence of factual errors in each sentence. Completeness and conciseness are evaluated through a key fact alignment task, which involves matching key facts with summary sentences. Conciseness is evaluated by verifying whether each sentence contains the essential key facts, while completeness is measured by determining whether every key fact is included in the summary. For the evaluation, we provide the key facts in UniSumEval. The calculation of the human scores across three dimensions follows the formulations in Appendix B.

**Annotation Qualification and Compensation.** We recruit three postgraduate students with proficient English skills to assess the quality of revised summaries. Only annotator who pass an English proficiency test—which simulates the evaluation of sentence-level errors for assessing faithfulness and verifying that key facts are included—are hired. Annotators receive compensation exceeding the U.S. minimum wage.

**Human Evaluation Results.** The inter-annotator agreement was moderate for both tasks, with Krippendorff's $\alpha$ of 0.45 for fact verification and 0.63 for key fact alignment. Table 13 shows human evaluation results across three dimensions. ReFeed achieves significant improvements, aligning with the findings highlighted in Section 5.2. While other pipelines show only marginal gains approaching zero in either faithfulness or conciseness, failing to achieve a balanced performance across all dimensions, ReFeed successfully overcomes these shortcomings. The strong alignment between automated evaluators and human evaluation scores provides compelling evidence for the effectiveness of reflective reasoning.

# I   Additional Experiment Details

## I.1   Training Details

We trained LLaMA-3.1-8B-Instruct on reasoning data generated using the P4's prompt, with QwQ-32B-preview as the teacher model. The training focused on making the model more handling feedback through receptive reasoning, excluding explicit goals and guidelines for reflective reasoning. The reasoning generation prompt was modified to align with the model's inherent reasoning structure as seen in Table 27. We follow the same quality control process as SumFeed-CoT and also shuffle the dimension order. The train settings and formats are the same as those in Section 4.2.

## I.2   Training Data Size

Table 14 presents statistics on the data sizes resulting from the two data construction processes. While the initial <document, summary, feedback> triplets are the same, ReFeed achieves more successful reasoning, ultimately retaining approximately 8K instances, whereas P4 retains only 5K. This result suggests that reflective reasoning exhibits a higher success rate in reasoning compared to P4.

| Pipeline | Reasoning Strategies | Feedback | Original Data | Reasoning Format Filtering | Verification-based Filtering | Ratio of Successful Refinement |
|---|---|---|---|---|---|---|
| P4 | Receptive | LLaMA 3.1-70B (High) | 14,505 | 7,382 | 2,806 | 38.01% |
| | | LLaMA 3.1-8B (Low) | 14,496 | 7,860 | 2,510 | 31.93% |
| ReFeed | Reflective | LLaMA 3.1-70B (High) | 14,505 | 9,179 | 3,922 | 42.73% |
| | | LLaMA 3.1-8B (Low) | 14,496 | 9,066 | 3,791 | 41.82% |

Table 14: Comparison of the data construction process between P4 and ReFeed. The two feedback generation models use checkpoints obtained through instruction tuning.

### I.3 Inference Latency

Table 15 demonstrates that ReFeed not only reaches performance levels comparable to the teacher model, QwQ-32B-preview, but also does so with significantly enhanced computational efficiency. Under conditions of high-quality feedback, ReFeed achieves composite scores nearly equivalent to those of the teacher model while achieving inference speeds approximately 4× faster. Moreover, when compared against alternative pipelines such as P4, Our approach offers a robustness advantage in terms of feedback quality.

### I.4 Comparison of Receptive and Reflective Reasoning

To examine the observation in Section 5.5, we perform a qualitative analysis of two distinct reasoning policies. Table 28-29 illustrates reasoning examples under receptive (P4) and reflective reasoning (ReFeed). Our analysis reveals that reflective reasoning demonstrates superior robustness in handling inaccurate feedback. For instance, as shown in Table 28, uncritically accepting flawed feedback on conciseness may undermine completeness—potentially resulting in the complete removal of a sentence. In contrast, ReFeed evaluates and filters such feedback, thereby achieving well-rounded improvements in multiple dimensions while enhancing completeness.

| Model | Pipeline | Composite | | Inference Time |
|---|---|---|---|---|
| | | High | Low | |
| L3.1 8B | $P4_{4k}$ | 55.3[*] | 54.3[*] | 54.5s |
| | $ReFeed_{4k}$ | 57.1[*] | 55.7[*] | 53.9s |
| | $ReFeed_{8k}$ | **57.8**[*] | **56.5**[*] | 40.0s |
| QwQ 32B (Teacher) | | 58.1[*] | 57.3[*] | 196.6s |

Table 15: Comparison of reflective reasoning over receptive reasoning on low- and high-quality feedback setups as . $P4_{FT}$ indicates P4 fine-tuned with 4K data. 4K and 8K refer to training data size. For inference time, we use a batch size of 1 on two NVIDIA L40S GPUs.

### I.5 Comparison with Direct Refinement Approach

We compare ReFeed with the "Direct Refinement" approach, which uses the same LLaMA-3.1 8B backbone but is trained to generate the revised summary directly from the original summary. This approach excludes any detection or reasoning modules and is designed to take the original summary as input, producing a revised summary in a single step, as specified in the prompt shown in Table 30.

As shown in Table 16, while Direct Refinement boosts faithfulness by 7.0 with little change in other metrics, showing gains of only 0.5 in completeness and 3.9 in conciseness, producing a limited average of 3.9,

| Pipeline | Faith. | Comp. | Conc. | Avg. |
|---|---|---|---|---|
| Before Refine | 78.0 | 46.4 | 76.4 | 66.9 |
| DR | 85.0 (+7.0) | 46.9 (+0.5) | 80.3 (+3.9) | 70.8 (+3.9) |
| **ReFeed** | **82.7** (+4.7) | **60.0** (+13.6) | **83.4** (+7.0) | **75.3** (+8.4) |

Table 16: Comparison between Direct Refinement (DR) and ReFeed. Faith., Comp., and Conc. denote Faithfulness, Completeness, and Conciseness, respectively.

ReFeed shows balanced improvements across all dimensions, more than doubling the average score to 8.4. This improvement can be attributed to the ReFeed's reflective reasoning process, which processes explicit feedback across multiple dimensions, selectively integrates it while filtering out noisy feedback with adverse effects, and employs backtracking to ensure balanced improvement, thereby achieving substantial gains across all dimensions rather than minor gains in isolated ones.

| Instruction for reasoning |
|---|
| - Faithfulness: reason about factual inconsistencies in the summary sentence.
- Completeness: reason about why the summary is each missing key content.
- Conciseness: reason about why the summary does not contain key content and contains unnecessary details. |
| **User Prompt** |
| Your task is to reason about the provided feedback and to refine the summary based on the provided feedback.

**Instruction**:
1. Give reasons about the provided feedback by considering the relevant dimension and provide a suggested fix to the summary:
{Instruction for reasoning on the target dimension (single)}
2. Revise the summary by incorporating the feedback.
3. Provide your response in the following format:
" '
Feedback Reasoning:
[Your reasoning on feedback and suggested fix]
Revised Summary:
[Your revised summary]
"'

**Document:**
{Document}

**Summary:**
{Summary}

**Feedback:**
{Feedback} |

Table 17: Prompt for P1 and P2. For P1, a prompt is sent once for a single target dimension, whereas in P2, a prompt is sent three times for each target dimension. In P2, "Summary" in inputs can be revised summary in previous step. The prompt includes a single reasoning instruction specific to the target dimension among the three dimensions.

---

**Instruction for Reasoning**

---

- Faithfulness: reason about factual inconsistencies in the summary sentence.
- Completeness: reason about why the summary is each missing key content.
- Conciseness: reason about why the summary does not contain key content and contains unnecessary details.

---

**User Prompt (Turn 1)**

---

Your task is to reason about the provided feedback and to refine the summary based on the provided feedback.

**Instruction**:
1. Give reasons about the provided feedback by considering the relevant dimension and provide a suggested fix to the summary:

{Instruction for reasoning on the dimension A}

{Instruction for reasoning on the dimension B}

{Instruction for reasoning on the dimension C}
2. Revise the summary by incorporating the feedback.
3. Provide your response in the following format:
" '
Feedback Reasoning:
[Your reasoning on feedback and suggested fix]
Revised Summary:
[Your revised summary]
" '
**Document:**

{Document}

**Summary:**

{Summary}

**Feedback:**

{Feedback on the dimension A}

---

**User Prompt (Turn N)**

---

Refine your refined summary again based on the provided feedback. Critique and Refine again based on the provided feedback.

**Feedback:**

{Feedback on Nth dimension}

---

Table 18: Prompt for P3. In Turn 1, instruction for reasoning on three dimensions are provided. From the next turn, only feedback is given.

---

**Instruction for Reasoning**

---

- Faithfulness: reason about factual inconsistencies in the summary sentence.
- Completeness: reason about why the summary is each missing key content.
- Conciseness: reason about why the summary does not contain key content and contains unnecessary details.

---

**User Prompt**

---

Your task is to reason about the provided feedback and to refine the summary based on the provided feedback.

**Instruction**:
1. Give reasons about the provided feedback by considering the relevant dimension and provide a suggested fix to the summary:
{Instruction for reasoning on dimension A}
{Instruction for reasoning on dimension B}
{Instruction for reasoning on dimension C}
2. Revise the summary by incorporating the feedback.
3. Provide your response in the following format:
" '
Feedback Reasoning:
[Your reasoning on feedback and suggested fix]
Revised Summary:
[Your revised summary]
" '
**Document:**
{Document}

**Summary:**
{Summary}

**Feedback:**
{Feedback on dimension A}
{Feedback on dimension B}
{Feedback on dimension C}

---

Table 19: Prompt of P4. The prompt includes all reasoning instructions, with their order aligned to the sequence of dimensions in the feedback.

---

**User Prompt (Reasoning Model)**

---

I summarized the following document:
{Document}
Summary of the above document:
{Summary}

Reason about the factually inconsistent span in the sentence. A span is factually inconsistent if it cannot be substantiated by the document. Give reasons for the factual inconsistency, point to the error span by stating "The error span:  and end your answer with a suggested fix to the summary.

---

**User Prompt (Refinement Model)**

---

Your task is to refine the summary based on the provided feedback.

**Instruction**:
1. Revise the summary by incorporating the feedback.
2. Provide your response in the following format:
" '
Revised Summary:
[Your revised summary]
" '

**Document:**
{Document}

**Summary:**
{Summary}

**Feedback:**
{Feedback}

---

Table 20: Prompt of DCR. This pipeline separates the reasoning and refinement modules. First, the reasoning module generates a critique based on detected labels. Then, the output from the reasoning module is used as the feedback input for the refinement model.

---

**User Prompt**

---

Your task is to refine the summary based on the provided feedback.

**Instruction**:
1. Revise the summary by incorporating the feedback.
2. Provide your response in the following format:
" '
Revised Summary:
[Your revised summary]
" '

**Document:**
{Document}

**Summary:**
{Summary}

**Feedback:**
The summary is not consistent with the source text. The source text does not mention the following facts:
{Incorrect Atomic Fact 1}
{Incorrect Atomic Fact 2}
...
{Incorrect Atomic Fact N}

The summary should not include information that is not present in the article. Please check the document for the correct information and make appropriate edits.

---

Table 21: Prompt for ACUEval. ACUEval uses detection labels at the atomic fact level. We follow the detection prompt from Wan et al. (2024) and adjust the refinement prompt to ensure compatibility with other pipelines.

| | |
|---|---|
| Fact Check | You will receive a document followed by a corresponding summary.
Your task is to assess the factuality of each summary sentence across nine categories:
* no error: the statement aligns explicitly with the content of the document and is factually consistent with it.
* out-of-context error: the statement contains information not present in the document.
* entity error: the primary arguments (or their attributes) of the predicate are wrong.
* predicate error: the predicate in the summary statement is inconsistent with the document.
* circumstantial error: the additional information (like location or time) specifying the circumstance around a predicate is wrong.
* grammatical error: the grammar of the sentence is so wrong that it becomes meaningless.
* coreference error: a pronoun or reference with wrong or non-existing antecedent.
* linking error: error in how multiple statements are linked together in the discourse (for example temporal ordering or causal link).
* other error: the statement contains any factuality error which is not defined here.

**Instruction:**
First, compare each summary sentence with the document.
Second, provide a single sentence explaining which factuality error the sentence has.
Third, answer the classified error category for each sentence in the summary.

Provide your answer in JSON format. The answer should be a list of dictionaries whose keys are "sentence", "reason", and "category":
["sentence": "first sentence", "reason": "your reason", "category": "no error", "sentence": "second sentence", "reason": "your reason", "category": "out-of-context error", "sentence": "third sentence", "reason": "your reason", "category": "entity error",]

**Document:**
{document}

**Summary with** {# of sentences} **sentences:**
{sentences}

**JSON Output:** |

Table 22: Prompt of the FineSurE for fact-checking tasks.

| | |
|---|---|
| Key Fact Extraction | You will be provided with a transcript. Your task is to decompose the summary into a set of "key facts".
A "key fact" is a single fact written as briefly and clearly as possible, encompassing at most 2-3 entities.

Here are nine examples of key facts to illustrate the desired level of granularity:
* Kevin Carr set off on his journey from Haytor.
* Kevin Carr set off on his journey from Dartmoor.
* Kevin Carr set off on his journey in July 2013.
* Kevin Carr is less than 24 hours away from completing his trip.
* Kevin Carr ran around the world unsupported.
* Kevin Carr ran with his tent.
* Kevin Carr is set to break the previous record.
* Kevin Carr is set to break the record by 24 hours.

**Instruction**:
First, read the summary carefully. Second, decompose the summary into (at most 16) key facts.

Provide your answer in JSON format. The answer should be a dictionary with the key "key facts" containing the key facts as a list:
{"key facts": ["first key fact", "second key facts", "third key facts"]}

**Summary**: {summary}

**JSON Output:** |
| Key Fact Alignment | You will receive a summary and a set of key facts for the same document.
Your task is to assess if each key fact is inferred from the summary.

**Instruction**:
First, compare each key fact with the summary.
Second, check if the key fact is inferred from the summary and then response "Yes" or "No" for each key fact. If "Yes", specify the line number(s) of the summary sentence(s) relevant to each key fact.

Provide your answer in JSON format. The answer should be a list of dictionaries whose keys are "key fact", "response", and "line number":
["key fact": "first key fact", "response": "Yes", "line number": [1], "key fact": "second key fact", "response": "No", "line number": [], "key fact": "third key fact", "response": "Yes", "line number": [1, 2, 3]]

**Summary**:
{summary}

{# of key facts} **key facts:**
{key facts}

**JSON Output:** |

Table 23: Prompt of the FineSurE for key fact extraction and key fact alignment tasks.

| Instruction for Reasoning |
| --- |
| - Faithfulness: Does this feedback accurately identify summary sentences?
* Out-of-article Error: Facts, new information or subjective opinions not found or verifiable by the document.
* Entity Error: Incorrect or misreferenced details about key entities such as names, dates, locations, numbers, pronouns, and events in the summary.
* Relation Error: Misrepresented relationships, such as incorrect use of verbs, prepositions, and adjectives.
* Sentence Error: the entire sentence entirely contradicts the information in the document.
- Completeness: Does this feedback correctly identify missing key content in the summary?
- Conciseness: Does the feedback correctly identify sentences that include unnecessary details and lack key content? |

| User Prompt |
| --- |
| Your goal is to deliberate on the provided feedback and propose actionable and specific aggregated feedback based on it.
Provide your response in the following format:
'''
**Final Reviesed Summary**:
\\[ \\boxed{\\text{Your revised summary}}\] 
'''

**Instruction**:
1. Deliberate on the characteristics an ideal summary should achieve.
2. Assess and choose the validity of the given feedback in improving the summary considering feedback quality criteria:
{Instruction for reasoning on dimension A}
{Instruction for reasoning on dimension B}
{Instruction for reasoning on dimension C}
3. Aggregate the valid feedback and Revise summary by incorporating it.
4. Carefully check whether each feedback and suggestion compromise other quality dimensions. Backtrack your reasoning If you need to.

**Document:**
{Document}

**Summary:**
{Summary}

**Ideal Summary:**
{Best Summary}

**Feedback:**
{Feedback on dimension A}
{Feedback on dimension B}
{Feedback on dimension C} |

Table 24: Reflective reasoning prompt for generating reasoning data in SumFeed-CoT (ReFeed)

| System | <\|begin_of_text\|><\|start_header_id\|>system<\|end_header_id\|> Your role as an assistant involves thoroughly exploring questions through a systematic long thinking process before providing the final precise and accurate solutions...<\|eot_id\|> |
|---|---|
| Input (User) | <\|start_header_id\|>user<\|end_header_id\|> Your goal is to deliberate on the provided feedback and propose actionable and specific aggregated feedback based on it.
Instructions:
1. Deliberate on the characteristics an ideal summary should achieve.
2. Assess and choose the validity of the given feedback in improving the summary considering feedback quality criteria:
- Conciseness: Does the feedback correctly identify sentences that include unnecessary details and lack key content?
- Faithfulness: Does this feedback accurately identify summary sentences that align with one of the following four factual inconsistencies types?:
- Completeness: Does this feedback correctly identify missing key content in the summary?
3. Aggregate the valid feedback and Revise summary by incorporating it. if no revision is needed, just answer "no revision needed".
4. Carefully check whether each feedback and suggestion compromise other quality dimensions. Backtrack your reasoning If you need to.

Document:
Person1: Good Morning. What can I do for you? \n' Person2: Good Morning, I have a bad cough, and I want to see an internist. \n' Person1: Do you have a registration card? \n' Person2: No, I don't. I'm a tourist. \n' Person1: Then you need to register as a new patient. Can I have a look at you ID card? I need your name, age, occupation, address and contacting number. \n' Person2: Here you are. \n' Person1: Thanks. Which kind of medicine do you prefer, western medicine or traditional Chinese medicine? \n' Person2: I'd like to try traditional Chinese medicine. I heard it really works quite well if you know what you're doing. \n' Person1: All right. Yes, he does. The registration fee is 7 yuan, please.

Summary:
A tourist asks to see an internist because he has a bad cough. The doctor asks for his name, age, occupation, address and contacting number. The tourist says he would like to try traditional Chinese medicine. The internist agrees to see him for a fee of 7 yuan.

Feedback:
***Conciseness Feedback***
These summary sentences do not contain key content:
Sentence 2: The doctor asks for his name, age, occupation, address and contacting number.
Sentence 4: The internist agrees to see him for a fee of 7 yuan.

***Faithfulness Feedback***
These summary sentences are factually inconsistent with the Document:
- Sentence 4: The internist agrees to see him for a fee of 7 yuan.

***Completeness Feedback***
These key contents are missing in the summary:
Missing key content 1: The registration fee is 7 yuan.
<\|eot_id\|> |
| Output (Assistant) | <\|begin_of_text\|><\|start_header_id\|>assistant<\|end_header_id\|>
<think>
Alright, I have this task to review a summary based on a given document and some feedback. I need to make sure the summary is accurate, complete, and concise. Let's break this down step by step.
First, I need to understand what an ideal summary should look like. A good summary should capture the main points of the document without including unnecessary details....

....

Now, let's look at the feedback provided:

Faithfulness Feedback:
- Sentence 4: "The internist agrees to see him for a fee of 7 yuan."
This feedback suggests that this sentence is factually inconsistent with the document. I need to check the document to see if this is accurate.

In the document:
- Person1 (likely a staff member) says, "The registration fee is 7 yuan, please."
There's no explicit statement that the internist agrees to see him for that fee. The fee seems to be for registration, not directly for the internist's service....

...

Let me try to revise the summary based on this analysis.
First, clarify the roles: It's likely that Person1 is a receptionist or a staff member, not the internist. The internist is probably the doctor the tourist wants to see....
...

</think>

<answer>
**Final Revised Summary:**
\[\boxed{\text{A tourist visits a medical facility with a bad cough and requests to see an internist. Since he doesn't have a registration card, he needs to register as a new patient. The staff member asks for his ID card to record his name, age, occupation, address, and contact number. The tourist expresses a preference for traditional Chinese medicine, and the staff informs him that the registration fee is 7 yuan.}} \]
</answer> <\|eot_id\|> |

Table 25: An example of an input (system prompt and user prompt) and its corresponding reasoning and revised summary (assistant prompt) used to train LLaMA-3.1-8b-instruct.

| Instruction for Reasoning |
|---|
| - Faithfulness: Does this feedback accurately identify summary sentences?
- Completeness: Does this feedback correctly identify missing key content in the summary?
- Conciseness: Does the feedback correctly identify sentences that include unnecessary details and lack key content? |

| System Prompt |
|---|
| Your role as an assistant involves thoroughly exploring questions through a systematic long thinking process before providing the final precise and accurate solutions. This requires engaging in a comprehensive cycle of analysis, summarizing, exploration, re-assessment, reflection, backtracing, and iteration to develop well-considered thinking process. Please structure your response into two main sections: Think and Answer. In the Think section, detail your reasoning process using the specified format: <think> {thought with steps separated with '\n \n'} </think> Each step should include detailed considerations such as analyzing questions, summarizing relevant findings, brainstorming new ideas, verifying the accuracy of the current steps, refining any errors, and revisiting previous steps. In the Answer section, based on various attempts, explorations, and reflections from the Think section, systematically present the final solution that you deem correct. The solution should remain a logical, accurate, concise expression style and detail necessary step needed to reach the conclusion, formatted as follows: <answer> {final formatted, precise, and clear solution} </answer> Now, try to solve the following question through the above guidelines: |

| User Prompt |
|---|
| Your goal is to deliberate on the provided feedback and propose actionable and specific aggregated feedback based on it.

**Instruction**:
1. Deliberate on the characteristics an ideal summary should achieve.
2. Assess and choose the validity of the given feedback in improving the summary considering feedback quality criteria:
{Instruction for reasoning on dimension A}
{Instruction for reasoning on dimension B}
{Instruction for reasoning on dimension C}
3. Aggregate the valid feedback and Revise summary by incorporating it.
4. Carefully check whether each feedback and suggestion compromise other quality dimensions. Backtrack your reasoning If you need to.

**Document:**
{Document}

**Summary:**
{Summary}

**Feedback:**
{Feedback on dimension A}
{Feedback on dimension B}
{Feedback on dimension C} |

Table 26: Prompt of ReFeed for inference. The prompt includes all reasoning instructions, tailored for reflective reasoning to validate noisy feedback, with their order aligned to the sequence of dimensions in the feedback.

| Instruction for Reasoning |
| --- |
| - Faithfulness: reason about factual inconsistencies in the summary sentence.
- Completeness: reason about why the summary is each missing key content.
- Conciseness: reason about why the summary does not contain key content and contains unnecessary details. |
| **User Prompt** |
| Your task is to reason about the provided feedback and to refine the summary based on the provided feedback.

Provide your response in the following format:
'''
**Final Reviesed Summary**:
\\[ \\boxed{\\text{Your revised summary}}\\]
'''
**Instruction**:
1. Give reasons about the provided feedback by considering the relevant dimension and provide a suggested fix to the summary:
{Instruction for reasoning on dimension A}
{Instruction for reasoning on dimension B}
{Instruction for reasoning on dimension C}
2. Revise the summary by incorporating the feedback.

**Document:**
{Document}

**Summary:**
{Summary}

**Feedback:**
{Feedback on dimension A}
{Feedback on dimension B}
{Feedback on dimension C} |

Table 27: Prompt for generating reasoning data through receptive reasoning (P4)

| Original Summary | Attorneys representing 8000+ women who were secretly recorded during pelvic exams by Dr. Nikita Levy at a Johns Hopkins-affiliated clinic will receive $32 million in legal fees from a $190 million settlement. The recordings were discovered in 2013, and Dr. Levy committed suicide ten days later. The settlement is one of the largest on record in the U.S. involving sexual misconduct by a physician. The case never produced criminal charges, but investigators found 1200 videos and 140 images stored on servers in his home. Dr. Levy's patients were traumatized, but their faces were not visible in the images and it could not be established which patients were recorded. Dr. Levy graduated from Weill Cornell Medical College and completed his internship and residency at Kings County Hospital Center. He began working at Hopkins in 1888 and saw roughly 12,600 patients during his 25-year tenure. The attorneys had originally asked for a 35% settlement fee, but a Baltimore judge ruled they would only receive 17.25%. The settlement closes a case that threatened the reputation of Johns Hopkins, one of the world's leading medical institutions. |
|---|---|
| Feedback (Ground truth) | Faithfulness Feedback : [0, 0, 0, 0, 0, 0, 1, 0, 0]
Completeness Feedback : [0, 0, 0, 0, 0, 1, 0, 0, 0, 0, 0, 1, 0, 1]
Conciseness Feedback : [0, 0, 0, 0, 0, 1, 0, 0, 0] |
| Feedback (Machine) | Faithfulness Feedback : [1, 1, 0, 1, 1, 0, 1, 0, 1]
Completeness Feedback : [0, 1, 0, 0, 0]
Conciseness Feedback : [0, 0, 0, 0, 1, 1, 1, 0, 1] |
| Original Scores (FineSurE) | Faithfulness Score : 89%
Completeness Score : 71%
Conciseness Score : 89% |
| P4 (Receptive reasoning) | Feedback Reasoning:

Faithfulness: The summary sentences are factually inconsistent with the Document. The date of Dr. Levy's start at Hopkins is incorrect (1888 instead of 1988), and the discovery of recordings was not in 2013, but it was discovered after a co-worker alerted hospital authorities, which led to his firing in February 2013. The sentence about the case never producing criminal charges is also inconsistent with the Document, as it mentions that investigators found 1200 videos and 140 images, but it does not explicitly state that the case never produced criminal charges.

Suggested fix: Revise the summary to reflect the correct information.

Completeness: The summary is missing key content, specifically the fact that Dr. Nikita Levy was fired in February 2013 after a co-worker alerted hospital authorities.

Suggested fix: Add this key content to the summary.

Conciseness: The summary contains unnecessary details, specifically sentences 5, 6, 7, and 9, which do not contain key content.

Suggested fix: Remove these sentences and focus on the essential information.

Revised Summary: Attorneys representing more than 8,000 women who were secretly recorded during pelvic exams by Dr. Nikita Levy at a Johns Hopkins-affiliated clinic will receive $32 million in legal fees from a $190 million settlement. Dr. Levy was fired in February 2013 after a co-worker alerted hospital authorities, and he committed suicide ten days later. The settlement is one of the largest on record in the U.S. involving sexual misconduct by a physician. Investigators found 1,200 videos and 140 images stored on servers in his home. The attorneys had originally asked for a 35% settlement fee, but a Baltimore judge ruled they would only receive 17.25%. |
| Revised Summary Scores | Faithfulness Score : 100%
Completeness Score : 57%
Conciseness Score : 100% |

Table 28: An example of refinement through receptive reasoning (P4). Red indicates incorrect feedback or flawed reasoning. Summary scores is evaluated by FineSurE

| | |
|---|---|
| Feedback (Machine) | Faithfulness Feedback : [1, 1, 0, 1, 1, 0, 1, 0, 1]
Completeness Feedback : [0, 1, 0, 0, 0]
Conciseness Feedback : [0, 0, 0, 0, 1, 1, 1, 0, 1] |
| P5 (Reflective reasoning) | \<think\> **Step 1: Understanding the Document**
First, I need to grasp the main points of the document. It's about a settlement involving a gynecologist, Dr. Nikita Levy, who secretly recorded and photographed thousands of women during pelvic exams at a Johns Hopkins-affiliated clinic. The settlement is for $190 million, with $32 million going to the attorneys representing over 8,000 women. Dr. Levy was fired in 2013 after a co-worker reported her suspicions, and he later committed suicide. The settlement is one of the largest for sexual misconduct by a physician in the U.S., and it closes a case that threatened the reputation of Johns Hopkins.

**Step 2: Evaluating Faithfulness**
I need to check if these sentences are factually inconsistent with the document.
- Sentence 1: Correct. It accurately states that attorneys representing over 8,000 women will receive $32 million from a $190 million settlement.
- Sentence 2: Correct. It states that the recordings were discovered in 2013 and that Dr. Levy committed suicide ten days later.
- Sentence 4: Correct. It mentions that the case never produced criminal charges and that investigators found 1,200 videos and 140 images on servers in his home.
- Sentence 5: Correct. It states that Dr. Levy's patients were traumatized, but their faces were not visible in the images, and it couldn't be established which patients were recorded.
- Sentence 7: Incorrect. According to the document, Dr. Levy began working at Hopkins in 1988, not 1888. 1888 is in the 19th century, and it seems like a typo or an error in the summary.
- Sentence 9: Correct. It states that the settlement closes a case that threatened the reputation of Johns Hopkins, one of the world's leading medical institutions.
So, there's a factual inconsistency in sentence 7 regarding the year Dr. Levy started working at Hopkins.

**Step 3: Evaluating Completeness**
Looking at the completeness feedback:
- Missing key content 1: "The gynecologist, Dr. Nikita Levy, was fired in February 2013 after a co-worker alerted hospital authorities."
This is indeed a key piece of information that is missing from the summary. The document states that Dr. Levy was fired in February 2013 after a co-worker alerted hospital authorities, and then he committed suicide ten days later. This is important because it sets the timeline and explains why Dr. Levy was fired.

**Step 4: Evaluating Conciseness**
I need to assess if these sentences contain key content or if they are unnecessary details.
- Sentence 5: This sentence is important because it mentions that Dr. Levy's patients were traumatized, but their faces were not visible in the images, and it couldn't be established which patients were recorded. This provides context on the impact of Dr. Levy's actions.
- Sentence 6: This sentence provides Dr. Levy's educational background and where he completed his internship and residency. While it is factual, it might not be essential for understanding the main points of the summary.
- Sentence 7: This sentence is incorrect due to the year being 1888 instead of 1988. It also mentions the number of patients and tenure, which could be considered key content.
- Sentence 9: This sentence is important because it highlights the impact of the settlement on Johns Hopkins' reputation.
Given that sentence 7 contains an error and might not be necessary, and sentence 6 is factual but not essential.

**Step 5: Revising the Summary**
Considering the feedback, I need to correct the year Dr. Levy started working at Hopkins, include the missing key content about Dr. Levy's termination, and ensure that the summary remains concise and accurate.
First, correct the year: Instead of "He began working at Hopkins in 1888," it should be "He began working at Hopkins in 1988."
Second, include the missing key content: "The gynecologist, Dr. Nikita Levy, was fired in February 2013 after a co-worker alerted hospital authorities."
Third, consider the conciseness of sentences. Sentence 6 provides Dr. Levy's educational background, which might not be essential for the summary. Sentence 7 contains an error and might be unnecessary.
Fourth, ensure that all key elements are included. \</think\>
\<answer\> **Final Revised Summary:**
\[\boxed{\text{Attorneys representing over 8,000 women who were secretly recorded during pelvic exams by Dr. Nikita Levy at a Johns Hopkins-affiliated clinic will receive $32 million in legal fees from a $190 million settlement. Dr. Levy was fired in February 2013 after a co-worker alerted hospital authorities about her suspicions, and he later committed suicide. The settlement is one of the largest on record in the U.S. involving sexual misconduct by a physician. Investigators found 1,200 videos and 140 images stored on servers in his home, but no evidence that he shared the material with others. He began working at Hopkins in 1988 and saw roughly 12,600 patients during his 25-year tenure. The attorneys had originally asked for a 35% settlement fee, but a Baltimore judge ruled they would only receive 17.25%. The settlement closes a case that threatened the reputation of Johns Hopkins, one of the world's leading medical institutions. }} \] \</answer\> |
| Revised Summary Scores (FineSurE) | Faithfulness Score : 100%
Completeness Score : 79%
Conciseness Score : 100% |

Table 29: An example of refinement through reflective reasoning (ReFeed). The original summary and summary scores are identical in Table 28. Red indicates incorrect feedback or flawed reasoning, while Blue highlights the proper validation of feedback, ensuring that only incorrect feedback is filtered out.

---

**User Prompt**

---

Your task is to refine the summary.

**Instruction**:
1. Revise the summary by considering the relevant dimension:
- Faithfulness: Correct factual inconsistencies in the summary.
- Completeness: Add any missing key information that is essential to the context of the summary.
- Conciseness: Revise summary to clearly and efficiently convey key content without unnecessary details.

**Document:**
{Document}

**Summary:**
{Initial Summary}

---

Table 30: Prompt for training the Direct Refinement Approach. The output uses the revised summary in SumFeed-CoT.

