# OpenReview forum: "ReFeed: Multi-dimensional Summarization Refinement with Reflective Reasoning on Feedback"
_colmweb.org/COLM/2025/Conference — COLM 2025_

### Official Review · Reviewer_7Nyd · 2025-05-10

**Rating:** 6
**Confidence:** 3
**Ethics Flag:** 1

**Summary:**

This paper proposes ReFeed, a refinement pipeline designed to tackle the challenges of addressing multiple quality dimensions in summarization, specifically faithfulness, completeness, and conciseness. The pipeline performs post-hoc refinement to manage trade-offs across these dimensions, accounting for issues such as ordering bias (sequential vs. simultaneous handling of dimensions) and deficiencies like low accuracy or insufficient robustness.
The paper investigates whether improving one quality dimension compromises others, and whether their pipeline can outperform not only pipelines that address a single dimension but also baselines that target multiple dimensions under various settings.
ReFeed uses a newly developed dataset that trains models to enhance summaries through reflective reasoning on feedback. The dataset is constructed in three stages: Goal Specification, Guideline Formulation, and Quality Control. It is used to fine-tune LLaMA-3.1-8B-Instruct with LoRA and DeepSpeed, enabling the model to generate reasoning and refined summaries.
ReFeed is evaluated against four baselines on the UniSumEval dataset, using feedback generated by FineSurE to guide refinement across diverse domains. The refined summaries are assessed for faithfulness, completeness, and conciseness using FineSurE with GPT-4o. A small human evaluation study is also conducted.
The results show that ReFeed outperforms the four baselines in terms of average summary quality, demonstrating excellent handling of the three target dimensions. The authors also highlight ReFeed’s robustness to ordering bias and variability in feedback quality.

**Questions To Authors:**

- Did you control for variables that may affect the quality dimensions? For example, the length of the summary output; I checked the example in the appendices and noticed that the summary produced by ReFeed is around 150 tokens, while the one generated by P4 is only about 105 tokens.
- What is the rationale behind the specific threshold values used during filtering? For instance, why were thresholds such as faithfulness = 1, completeness ≥ 0.5, and conciseness ≥ 0.5 chosen?
- Did you attempt to optimize the prompts used in the pipelines, for example, by rephrasing, adding more information, or experimenting with different prompt structures?

**Reasons To Accept:**

- The paper addresses important challenges in multi-dimensional summarization, and the proposed pipeline appears to outperform several baselines, particularly in managing multiple quality dimensions with reasonable trade-offs.
- The dataset developed in the paper could be useful to the research community and may be reused or extended by other researchers.
- The paper is reasonably well-presented; the experiments consider different baselines, and most design choices regarding the pipeline components and evaluation strategy are well-justified.

**Reasons To Reject:**

- While the paper is generally well-structured, the presentation is quite dense, especially in its descriptions of the pipeline and dataset construction. Several important details are only fully explained in the appendix, making it difficult to follow without frequent back-and-forth reading.
- The evaluation relies heavily on automatic metrics, primarily using FineSurE with GPT-4o. While this is acceptable, I think the paper would benefit from a deeper error analysis and more extensive manual inspection of the refined summaries.
- While evaluating the end-to-end performance of the pipelines is expected and helpful, the paper would benefit from evaluating intermediate components or substeps, even partially through manual inspection. For example, checking the low- and high-quality feedback or analyzing samples from the <document, summary, feedback> triplets collected during the Initial Feedback Collection could provide deeper insight and help understand the impact of each step/component.

---

> ### Author Response · Authors · 2025-06-03
> **Response to Reviewer 7Nyd (1/4)**
>
> We appreciate your valuable comments and suggestions. We hope that the concerns can be resolved through our clarifications in this response.
>
> ---
>
> > **Weakness 1.** *While the paper is generally well-structured, the presentation is quite dense, especially in its descriptions of the pipeline and dataset construction. Several important details are only fully explained in the appendix, making it difficult to follow without frequent back-and-forth reading.*
>
>
> Thank you for the feedback. We agree that important details were moved to the appendix due to space limits. As the camera ready version allows an extra page, we plan to move this content to the main paper to improve clarity.
>
> ---
>
> > **Weakness 2.** *The evaluation relies heavily on automatic metrics, primarily using FineSurE with GPT-4o. While this is acceptable, I think the paper would benefit from a deeper error analysis and more extensive manual inspection of the refined summaries.*
>
> While our evaluation does indeed rely on automatic metrics, we would like to clarify that we also conducted manual inspection (human evaluation) of the refined summaries (in Table 13) on a subset of the results to complement the automatic evaluation metrics.
>
> The table below shows the human evaluation results compared alongside FineSurE scores:
>
> | Pipelines     | Faithfulness (Human) | Faithfulness (FineSurE) | Completeness (Human) | Completeness (FineSurE) | Conciseness (Human) | Conciseness (FineSurE) |
> | ------------- | ------------- | ---------------- | ------------ | --------------- | ------------ | --------------- |
> | `Before Refine` | 77.4          | 78.0             | 42.8         | 46.4            | 80.6         | 76.4            |
> | `P2`            | 77.3          | 78.4             | 45.8         | 51.5            | 84.4         | 84.8            |
> | `P3`            | 78.0          | 78.9             | 50.8         | 53.2            | 79.3         | 80.0            |
> | `P4`            | 84.5          | 80.1             | 52.0         | 56.0            | 81.1         | 83.6            |
> | `ReFeed`        | 87.3          | 82.7             | 54.2         | 60.0            | 88.7         | 83.4            |
>
> As the table indicates, although there are some absolute differences between human scores and automatic metrics, the relative performance trends across different pipelines are consistent.
>
> We plan to expand this analysis in the paper by providing a detailed breakdown of error types before and after refinement.

---

> ### Author Response · Authors · 2025-06-03
> **Response to Reviewer 7Nyd (2/4)**
>
> > **Weakness 3.** *While evaluating the end-to-end performance of the pipelines is expected and helpful, the paper would benefit from evaluating intermediate components or substeps, even partially through manual inspection. For example, checking the low- and high-quality feedback or analyzing samples from the <document, summary, feedback> triplets collected during the Initial Feedback Collection could provide deeper insight and help understand the impact of each step/component.*
>
> We appreciate your insightful suggestion. We present an ablation study to evaluate the contribution of individual substeps in our data construction pipeline.
>
> **First, analysis on <document, summary, feedback> triplets:**
> A key design choice in our data construction was to include feedback with varying quality levels from "low" to "high" (`Original`), enhancing the model’s ability to filter out noisy signals through reflective reasoning. Thus, as part of our component analysis, we evaluated the effect of using only low-quality feedback (`Only Low`), simulating a constrained feedback diversity setting.
>
> The table below compares the summary quality after refinement by `ReFeed`, where one model is trained with mixed-quality feedback and the other with only low-quality feedback. The summary quality is evaluated using FineSurE (with GPT-4o). The `Only Low` setting underperforms compared to the original mixed-feedback configuration, yielding lower scores across all evaluation dimensions. Therefore, including diversified feedback in the training dataset is essential to equip the model with the ability to distinguish and utilize high-quality signals through reflective reasoning.
>
>
> | Configuration | Faithfulness | Completeness | Conciseness | Avg.        |
> | ------------- | ------------ | ------------ | ----------- | ----------- |
> |`Original`      | 82.7 (+4.7)  | 60.0 (+13.6) | 83.4 (+7.0) | 75.3 (+8.4) |
> | `Only Low`      | 81.3 (+3.3)  | 55.1 (+8.7)  | 82.8 (+6.4) | 73.1 (+6.2) |
>
> **Second, analysis on filtering in training dataset:**
> As described in Lines 221–230, we applied filtering to remove malformed formats and failed refinement cases for high-quality reasoning data. While this substep improves data quality, it also reduces dataset size (16K → 8K), which is a key component of SFT. We examined the necessity of this substep within the training pipeline.
>
> From the table below, although `w/o Filtering` increased data volume, it resulted in limited performance in conciseness (83.4 → 80.2). This finding suggests that high quality reasoning, though sparser, is essential for balanced multi-dimensional gains. While we currently rely on single sampling from QwQ, we believe performance could be further improved by gathering more high-quality feedback through iterative sampling.
>
> | Configuration | Faithfulness | Completeness | Conciseness | Avg.        |
> | ------------- | ------------ | ------------ | ----------- | ----------- |
> | `Original`      | 82.7 (+4.7)  | 60.0 (+13.6) | 83.4 (+7.0) | 75.3 (+8.4) |
> | `w/o Filtering`         | 83.5 (+5.5)  | 61.1 (+14.7) | 80.2 (+3.8) | 75.3 (+8.4)  |
>
> We will incorporate this ablation into the revised version of the paper to address your suggestion.

---

> ### Author Response · Authors · 2025-06-03
> **Response to Reviewer 7Nyd (3/4)**
>
> > **Qustion 1.** *Did you control for variables that may affect the quality dimensions? For example, the length of the summary output; I checked the example in the appendices and noticed that the summary produced by ReFeed is around 150 tokens, while the one generated by P4 is only about 105 tokens.*
>
> We did not explicitly control variables such as summary length or other factors like abstractiveness. This is because the goal of refinement in our paper is not to produce outputs with fixed properties, but rather to improve summary quality based on feedback across multi-dimensions.
>
> Since these variables are tied with the dimensions being optimized,  controlling such variables would not only limit the model’s flexibility in adapting to feedback. Specifically, variables such as summary length are inherently outcomes of the refinement for multi-dimensions. These variables vary depending on which quality dimension is emphasized during refinement.
>
> For instance:
>
> - When `ReFeed` summary is shorter than `P4` (`ReFeed < P4`), The improvement is often due to better handling of verbosity, leading to increased conciseness.
> - Conversely, when `ReFeed` summary is longer than `P4` (`ReFeed > P4`), the increased summary length reflects gains in completeness, as more key information is incorporated in summary.
>
> This pattern is supported by the results shown in the table below, where Δ indicates the change in a given dimension before and after refinement.
>
> | Case | ΔCompleteness (P4) |  ΔCompleteness (ReFeed) |  ΔConciseness (P4) |  ΔConciseness (ReFeed) |
> |------------|---------------------|---------------------|--------------------|--------------------|
> | `ReFeed < P4`    | +15.9                | +17.6                | +7.7                | +8.2                |
> | `ReFeed > P4`    | -0.7                | +7.0                 | +6.4                | +6.2                |
>
> As shown, in cases where `ReFeed < P4`, ReFeed achieves greater conciseness gains than P4 (8.2 vs. 7.7), effectively eliminating unnecessary details. In contrast, when `ReFeed > P4`, ReFeed shows a substantially larger completeness improvement over P4 (7.0 vs. -0.7) compared to the smaller gap observed in the `ReFeed < P4` setting (17.6 vs. 15.9). This improvement comes with a slight reduction in conciseness gain (6.2 vs. 6.4), representing a reasonable trade-off when completeness is prioritized.
>
> Overall, this analysis suggests that summary length or similar variables are best treated as emergent properties of the refinement process, rather than fixed constraints. As such, we believe that these variables should remain unconstrained. We clarify this in our revised manuscript.
>
> ---
>
>
> > **Qustion 2.** *What is the rationale behind the specific threshold values used during filtering? For instance, why were thresholds such as faithfulness = 1, completeness ≥ 0.5, and conciseness ≥ 0.5 chosen?*
>
> Each threshold is derived from observations reported in the UniSumEval paper. The table below shows the corresponding average scores for faithfulness, completeness, and conciseness.
>
> | Document Context     | Faithfulness | Completeness | Conciseness |
> | -------------------- | ---------------- | ---------------- | --------------- |
> | `Non-Dialogue (Short)`| 92.1             | 67.0             | 86.4            |
> | `Non-Dialogue (Long)`  | 93.1             | 31.7             | 67.7            |
> | `Dialogue (Short)`    | 91.0             | 56.0             | 77.7            |
> | `Dialogue (Long)`     | 88.0             | 44.8             | 73.1            |
>
> As shown in the table, faithfulness is consistently high (around 0.9), but completeness and conciseness vary more by document context (length, type and domain), making 1.0 an unrealistic target. Hence, we adopt a threshold of 1.0 for faithfulness, and 0.5 for completeness and conciseness as practical lower bounds to identify summaries of acceptable quality. We will clarify this in our revised manuscript.

---

> ### Author Response · Authors · 2025-06-03
> **Response to Reviewer 7Nyd (4/4)**
>
> > **Qustion 3. _Did you attempt to optimize the prompts used in the pipelines, for example, by rephrasing, adding more information, or experimenting with different prompt structures?_**
>
> We attempted to optimize the prompts used across all our pipelines, including P1–P4 and ReFeed. This led us to conclude that, as long as the same pipeline is followed, prompt engineering such as rephrasing or adding more information yields only minor improvements, and that major gains require a transformation in reasoning style induced through training.
>
> For example, we refined the prompt used in the P4 pipeline (see Table 18) as shown below. This revised prompt includes more detailed information to better address challenges such as trade-offs among multiple feedback signals.
> ```
> Your task is to reason about the provided feedback and refine the summary accordingly, while carefully balancing the relevant quality dimensions.
> Keep in mind that improvements in one dimension may affect others (e.g., increasing completeness could reduce conciseness).
> Be cautious about the possibility of biased or noisy feedback.
>
> Instruction:
> 1. Give reasons about the provided feedback by considering the relevant dimension and provide a suggested fix to the summary:
> - Faithfulness: reason about factual inconsistencies in the summary sentence.
> - Completeness: reason about why the summary is each missing key content.
> - Conciseness: reason about why the summary does not contain key content and contains unnecessary details.
> 2. When reasoning, consider possible trade-offs between dimensions and address them explicitly if they arise.
> 3. Assess whether the feedback might be noisy or biased, and explain how you choose to handle it.
> 4. Provide your response in the following format:
> """
> Feedback Reasoning:
> [Your reasoning on feedback and suggested fix]
>
> Revised Summary:
> [Your revised summary]
> """
>
> Document:
> {Document}
>
> Summary:
> {Initial Summary}
>
> Feedback:
> {Feedback}
> ```
>
> The table below contrasts the summary quailty after refinement using `P4` and `P4-Advanced` prompts, where the summary quality is evaluated by FineSurE using GPT-4o.
>
> |                  | Faithfulness | Completeness | Conciseness | Average     |
> | ---------------- | ------------ | ------------ | ----------- | ----------- |
> | `Before Refine`    | 78.0         | 46.4         | 76.4        | 66.9        |
> | `P4`               | 80.1 (+2.1)  | 56.0 (+9.6)  | 83.6 (+7.2) | 73.2 (+6.3) |
> | `P4-Advanced` | 81.3 (+3.3)  | 55.1 (+8.7)  | 82.8 (+6.4) | 73.1 (+6.2) |
> | `ReFeed`           | 82.7 (+4.7)  | 60.0 (+13.6) | 83.4 (+7.0) | 75.3 (+8.4) |
>
> We observe that there is marginal difference between `P4` and `P4-Advanced`; while the scores vary across individual dimensions, the overall average remains similar. Therefore, simply adding more information to the prompt is not sufficient to effectively handle multi-dimensional feedback, highlighting the necessity of reflective reasoning as employed in our `Refeed` framework.
>
> We will explore additional prompt variations and include the corresponding results and analysis in the revised manuscript.

---

> ### Author Response · Authors · 2025-06-07
> **Gentle reminder to review the response**
>
> Dear Reviewer 7Nyd
>
> We find your comments and feedback invaluable and have done our best to address them. Your suggestions have been incorporated into our revised manuscript. As the rebuttal period draws to a close, we kindly request your review of our response. Please do not hesitate to contact us if you need any further clarification or have additional queries.
>
> Sincerely, The authors

---

> ### Comment · Reviewer_7Nyd · 2025-06-09
>
> Thank you for your replies. I will keep my score as it is.

---

### Official Review · Reviewer_qsDY · 2025-05-12

**Rating:** 6
**Confidence:** 3
**Ethics Flag:** 1

**Summary:**

The paper proposes a summarization refinement pipeline that distills large reasoning models to perform summarization. The paper first constructs SumFeed-CoT that captures reflective reasoning of the LLMs while giving feedback about summaries along three dimensions ((faithfulness, completeness, conciseness)). Then, a smaller model is trained on this data to perform the reflective reasoning and refine the summary. Automatic metrics show that the proposed approach outperforms the existing summary refiners in terms of quality and robustness.

**Questions To Authors:**

Does UniSumEval evaluation dataset  include human-written reference summaries  to facilitate other reference-based automatic metrics?

**Reasons To Accept:**

New approach for refining summaries. New synthetic dataset to train/refine summarization systems.

The analysis section is comprehensive and thorough. The section on robustness is especially insightful.

**Reasons To Reject:**

Missing baselines:

1) All the baselines in Table 1  are trained on (document, summary, feedback) triples as input and (reasoning, refined summary) as output. Ideally there should be a baseline that is trained directly on the refined summaries in SumFeed-CoT to understand how much value reflective reasoning/feedback is adding.
2) Ideally, there should also be a couple of baselines that do not perform any distillation - such as using standard zero-shot or few shot prompting techniques.

---

> ### Author Response · Authors · 2025-06-03
> **Response to Reviewer qsDY (1/2)**
>
> We appreciate your valuable comments and suggestions. We hope that the concerns can be resolved through our clarifications in this response.
>
> ---
>
> > **Weakness 1.** *All the baselines in Table 1 are trained on (document, summary, feedback) triples as input and (reasoning, refined summary) as output. Ideally there should be a baseline that is trained directly on the refined summaries in SumFeed-CoT to understand how much value reflective reasoning/feedback is adding.*
>
> In response to your suggestion, we include an additional comparison against a baseline trained directly on the refined summaries from SumFeed-CoT, which we refer to as `Direct Refinement`. This baseline is trained using the prompt template provided below.
>
> ```
> Your task is to refine the summary.
>
> Instruction:
> 1. Revise the summary by considering the relevant dimension:
> - Faithfulness: Correct factual inconsistencies in the summary.
> - Completeness: Add any missing key information that is essential to the context of the summary.
> - Conciseness: Revise summary to clearly and efficiently convey key content without unnecessary details.
>
> Document:
> {Document}
>
> Summary:
> {Initial Summary}
> ```
>
> The table below compares the effectiveness of `ReFeed` with `Direct Refinement`. While `Direct Refinement` improves faithfulness, its performance on other dimensions is substantially lower than that of `ReFeed`. These results indicate that `Direct Refinement` tends to over-optimize for faithfulness while failing to address other dimensions effectively, due to the lack of explicit feedback and reasoning. Consequently, it yields a substantially lower average score compared to `ReFeed`.
>
>
> |         Pipeline            | Faithfulness       | Completeness        | Conciseness        | Average         |
> |---------------------|--------------|--------------|--------------|--------------|
> | `Before Refine`   | 78.0         | 46.4         | 76.4         | 66.9         |
> | `Direct Refinement`       | 85.0 (+7.0)  | 46.9 (+0.5)  | 80.3 (+3.9)  | 70.8 (+3.9)  |
> | `ReFeed`  | 82.7 (+4.7)  | 60.0 (+13.6) | 83.4 (+7.0)  | 75.3 (+8.4)  |
>
>
> We will incorporate `Direct Refinement` to better illustrate the benefits of feedback based refinement for multiple dimensions in our revised manuscript.
>
> ---

---

> ### Author Response · Authors · 2025-06-03
> **Response to Reviewer qsDY (2/2)**
>
> > **Weakness 2.** *Ideally, there should also be a couple of baselines that do not perform any distillation - such as using standard zero-shot or few shot prompting techniques.*
>
> In line with your suggestion, we include `Zero-shot` and `One-shot` variants of `ReFeed`, as shown in the table below.
>
> |    Pipeline                    | Faithfulness       | Completeness        | Conciseness        | Average         |
> |------------------------|--------------|--------------|--------------|--------------|
> | `Before Refine`          | 78.0         | 46.4         | 76.4         | 66.9         |
> | `P4`                     | 80.1 (+2.1)  | 56.0 (+9.6)  | 83.6 (+7.2)  | 73.2 (+6.3)  |
> | `ReFeed (Zero-shot)`           | 81.9 (+3.9)  | 57.1 (+10.7) | 65.1 (–11.3) | 68.0 (+1.1)  |
> | `ReFeed (One-shot)`| 73.2 (–4.8)  | 48.8 (+2.4)  | 62.5 (–13.9) | 61.5 (–5.4)  |
> | `ReFeed (Distillation)`          | 82.7 (+4.7)  | 60.0 (+13.6) | 83.4 (+7.0)  | 75.3 (+8.4)  |
>
>
> `Zero-shot` improves over `P4` in Faithfulness and Completeness but drops sharply in Conciseness, indicating a trade-off. Specifically, non-reasoning models without any distillation are limited in their ability to revisit earlier outputs to manage such trade-off, and they struggle to validate noisy feedback effectively.
>
> Moreover, the `One-shot` with example in SumFeed-CoT shows that reflective reasoning cannot be mimicked via a few examples. This often results in overthinking, where unnecessary revisions occur, leading to performance drop.
>
> This concludes that prompt engineering alone is insufficient for reflective reasoning, and emphasize the importance of model alignment through distillation with Long-CoT.
>
> ---
>
> > **Quesion 1.** *Does UniSumEval evaluation dataset include human-written reference summaries to facilitate other reference-based automatic metrics?*
>
> UniSumEval includes human-written reference summaries from each of the source datasets, not newly curated ones. However, many original reference summaries in UniSumEval are of relatively low quality, often lacking professional editing or rigorous quality control [1].
>
> Furthermore, summarization is an open-ended generation task, where multiple valid summaries may exist rather than a single optimal answer. Given this inherent diversity, reference-free evaluation has become a widely accepted primary approach, such as G-Eval [2] and FineSurE [3], which is an evaluation strategy we also follow in this work.
>
>
> [1] Zhang et al, Benchmarking Large Language Models for News Summarization, Transactions of the Association for Computational Linguistics, 2024
>
> [2] Liu et al, G-Eval: NLG Evaluation using GPT-4 with Better Human Alignment, In EMNLP, 2023
>
> [3] Song et al, FineSurE: Fine-grained Summarization Evaluation using LLMs, In ACL, 2023

---

> ### Author Response · Authors · 2025-06-07
> **Gentle reminder to review the response**
>
> Dear Reviewer qsDY
>
> We find your comments and feedback invaluable and have done our best to address them. Your suggestions have been incorporated into our revised manuscript. As the rebuttal period draws to a close, we kindly request your review of our response. Please do not hesitate to contact us if you need any further clarification or have additional queries.
>
> Sincerely, The authors

---

> ### Comment · Reviewer_qsDY · 2025-06-08
> **Reviewer response**
>
> Thank you  for the reply. I would still like to keep the same score.

---

### Official Review · Reviewer_iDgk · 2025-05-13

**Rating:** 7
**Confidence:** 3
**Ethics Flag:** 1

**Summary:**

The paper proposes a summarization system that seeks to jointly optimize for faithfulness, completeness, and conciseness.

The authors started by building a dataset (SummFeed-CoT) that pulls together LLM's reasoning from feedback, and trained a model on this dataset. They described 4 pipelines with varying levels of complexity (P1 to P4) in terms of how they leverage feedback. These 4 pipelines will go on to serve as baselines during subsequent evaluations. The authors then proposed ReFeed - which further builds on the most complex P4 pipeline, by also using reflective reasoning during refinement.

Experiments are conducted on the UniSumEval dataset, and FineSurE (LLM-based evaluator). The experiments are focused on 3 questions the authors posed, that 1/ looks at the overall effectiveness of pipelines P1 to P4 as well as ReFeed along the 3 key dimensions of faithfulness, completeness, and conciseness, 2/ the interplay between these 3 key dimensions and their effect on outcomes, and 3/ the impact due to the quality of the feedback received.

ReFeed is showmn to do better than the other baselines in terms of Faithfulness, and an average metric computed out of scores achieved for all 3 dimensions. The experiments also show that ReFeed is fairly robust against the order in which feedback is received.

**Questions To Authors:**

Suggestion #1 - Figure 1 shows both the construction of your dataset, as well as ReFeed. I understand editorial space is a premium, but for better readability, it could help to tease these out into two separate diagrams.

Suggestion #2 - While P1 and P4 serves as baseline for your evaluations, I also thought they might be distracting from the proposal you are making. If I understand the paper correctly, you are making a case for the use of better reflective reasoning to produce better summarizers that do well for faithfulness, completeness and conciseness. It will be more direct to explain ReFeed, and then benchmark it against other large reasoning models on your evaluation dataset. I could have missed the point behind it, but it's not clear what P1 and P4 add to the proposal here. [Update: Authors addressed this reasonably well]

Suggestion #3 - There is a lot of good discussion and elaboration in the authors' responses. I know it will not be easy, but hopefully you can incorporate many of the suggestions and additional results if space permits.

**Reasons To Accept:**

1. The dataset that comes with the paper would be interesting, and likely useful to the community.
2. The proposal builds on up-to-date datasets and evaluation metrics.
3. The experiments are well-designed to help build a case for the efficacy of the proposed ReFeed system. The authors have put in significant effort to run an extensive set of experiments.

**Reasons To Reject:**

1. It is hard to fully reproduce the human evaluations backing Table 1, so it is not possible to fully reason about the relative performance of ReFeed. [Update: The authors reproduction with Claude 2.1 comes close and helps close this gap. It could be useful to be more explicit about this in the paper.]
2. It is not clear what the motivations behind targeting reflecting reasoning is in this proposal. It would have been more compelling to make an argument of why this approach makes sense, and how it overcomes inherent weaknesses or challenges within current reasoning models.

---

> ### Author Response · Authors · 2025-06-03
> **Response to Reviewer iDgk (1/2)**
>
> We appreciate your valuable comments and suggestions. We hope that the concerns can be resolved through our clarifications in this response.
>
> ---
>
> > **Weakness 1.** *It is hard to reason about the relative performance of ReFeed given the lack of comparative studies against other summarization systems. I looked up the results of common baselines described in the UniSummEval paper. Admittedly I did not study all the figures in detail. However it would seem that the baselines score well in the 90s compared to what ReFeed (and P1 to P4) achieved in Table 1.*
>
> Thank you for pointing out the performance discrepancy related to UniSumEval. This misalignment stems from two factors (though the evaluation itself is not flawed):
> * UniSumEval is based on human evaluation, whereas our main results rely on LLM-based evaluation (human results are provided in Appendix H).
> * Human evaluations are inherently difficult to fully reproduce, which can lead to score discrepancies.
>
> Details on each point are provided below.
>
> **First, LLM-based evaluation:** The main results in Tables 1-5 were based on the latest LLM-based evaluation metric `FineSurE` [4] tailored for text summarization (using GPT-4o as its backbone). Given GPT-4o’s strong evaluation performance (75.9% BAcc) [5], we used the automated evaluator following recent summarization works [1, 2, 3, 4]. Nevertheless, we also conducted human evaluations and confirmed that they show consistent trends with the LLM-based results. These human evaluation results are included in Appendix H. Therefore, the discrepancy with UniSumEval arises from methodological differences, as our evaluation is based on GPT-4o (via `FineSurE`), while UniSumEval relies on human judgment.
>
> **Second, Challenges in Reproducing Human Judgments:** Since we also conducted human evaluations (in Appendix H), it is reasonable to compare these scores with those from UniSumEval. However, fully reproducing human evaluation results is inherently challenging due to annotator variability, subjective interpretation, and contextual ambiguity, especially when AI assistance is integrated into the annotation pipeline, as in the case of UniSumEval. Specifically, although we strictly followed the UniSumEval protocol in our human evaluation, we observed a notable discrepancy in the "Faithfulness" dimension, while the other two dimensions remained largely consistent.
>
> Our in-depth investigation during the rebuttal period revealed that this discrepancy stems not only from annotator variability, but also from the choice of AI assistance model, which significantly impacts the evaluation outcome. We observed that UniSumEval used Claude v2.1 as the AI assistant, whereas we used GPT-4o. Notably, GPT-4o tends to produce more conservative reasoning, frequently flagging content as potential hallucination compared to Claude v2.1. As a result, human annotators guided by GPT-4o were more likely to assign lower Faithfulness scores overall. However, using GPT-4o as the assistant is not problematic, as it has shown superior performance in recent automated evaluation benchmarks, justifying its use for supporting human evaluation: for fact checking, 75.9% BAcc with GPT-4o vs. 70.7% BAcc with Claude-v2.1 [5].
>
> Nevertheless, for completeness, we present an additional human evaluation using Claude v2.1, which is the same assistant model used by UniSumEval, and observed improved alignment with their reported scores. The table below presents human evaluation results across the three dimensions, using Claude v2.1 for AI assistance, consistent with the setup in the UniSumEval paper. These results allowed us to reproduce UniSumEval’s baseline Faithfulness score at a comparable level (~90%), acknowledging that exact replication is not possible due to annotator variability. Importantly, the results remain consistent with our main findings in Table 1, reaffirming that our proposed method achieves the strongest performance.
>
> | Metric       | Before Refine (Claude 2.1) | P2 (Claude 2.1) | P3 (Claude 2.1) | P4 (Claude 2.1) | ReFeed (Claude 2.1) |
> |--------------|-----------------------------|------------------|------------------|------------------|------------------|
> | `Faithfulness` | 88.2                        | 88.5             | 88.6             | 88.7             | 96.6             |
> | `Completeness` | 42.8                        | 45.8             | 50.8             | 52.0             | 54.2             |
> | `Conciseness`  | 80.6                        | 84.5             | 79.3             | 81.1             | 88.7   |
>
> Therefore, the misalignment between our results and those of UniSumEval does not indicate a flaw in our evaluation. Rather, it reflects the inherent difficulty of perfectly reproducing human evaluation results, as well as the fact that our main experimental results rely on LLM-based evaluation (with human evaluation reported only in the appendix).

---

> ### Author Response · Authors · 2025-06-03
> **Response to Reviewer iDgk (2/2)**
>
> > **Weakness 2.** *It is not clear what the motivations behind targeting reflecting reasoning is in this proposal. It would have been more compelling to make an argument of why this approach makes sense, and how it overcomes inherent weaknesses or challenges within current reasoning models.*
>
> We apologize for not clearly conveying our motivation. Our motivation stems from key limitations of prior refinement methods: they have all relied on "non-reasoning" models and, most notably, focused exclusively on single-dimension (primarily faithfulness). In other words, existing methods, such as DCR and ACUEval, operate under a receptive reasoning paradigm, passively accepting feedback as is. Consequently, they fall short of accounting for the multi-dimensional feedback we highlighted in Lines 31-34: (1) trade-off acorss dimensions; (2) ordering bias; and (3) noisy feedback.
>
> Our key idea lies in shifting from "receptive" to "reflective" reasoning, enabling models to move beyond passively accepting feedback and instead actively perform self-correction and self-verification through backtracking. This paradigm shift is not only conceptually well-founded but also empirically effective in handling the trade-offs and conflicts, such as order bias and noise, that emerge in multi-dimensional feedback.
>
> Accordingly, we position our work as the first to incorporate reflective reasoning into the domain of summarization refinement. In addition, we demonstrate that such reasoning capabilities can be effectively distilled into smaller models via our carefully constructed training dataset. We will clarify this more clearer in our revised manuscript.
>
> ---
>
> > **Suggestion 1.** *Figure 1 shows both the construction of your dataset, as well as ReFeed. I understand editorial space is a premium, but for better readability, it could help to tease these out into two separate diagrams.*
>
> Thank you for the suggestion. For better clarity, we will divide Figure 1 into two parts: the left part will detail the dataset construction process, while the right part will depict the reflective reasoning pipeline.
>
> ---
>
> > **Suggestion 2.** *While P1 and P4 serves as baseline for your evaluations, I also thought they might be distracting from the proposal you are making. If I understand the paper correctly, you are making a case for the use of better reflective reasoning to produce better summarizers that do well for faithfulness, completeness and conciseness. It will be more direct to explain ReFeed, and then benchmark it against other large reasoning models on your evaluation dataset. I could have missed the point behind it, but it's not clear what P1 and P4 add to the proposal here.*
>
> P1 and P4 highlight key challenges that have been largely overlooked in prior single dimensional and non-reasoning model setups. To our knowledge, no prior work has analyzed which prompt designs are necessary for multi dimensional refinement, nor how three key challenges influence performance across dimensions.
>
> At this point, our work moves beyond previous efforts by exploring diverse multi-dimensional refinement strategies. Building on these insights from P1-P4, ReFeed not only select reflective reasoning style but also introduces simultaneous refinement prompt design.
>
> Regarding comparisons with large-params LRMs, we would like to clarify that our goal is not to compete with such models. Instead, we align with the requirement in refinement tasks, where lightweight models are essential to meet high demand with fast and accurate corrections. Since LRMs are not suited for this purpose due to their scalability, we believe direct comparisons would be of limited relevance.
>
> Despite this, we provided the comparison against a large reasoning model, QwQ in Table 4, to show that our student models achieve comparable performance with lower latency. Specifically, ReFeed (8B) achieves scores of 57.8 and 56.5 under low- and high-quality feedback conditions, respectively, with a latency of 40 seconds. In contrast, QwQ (32B) shows slightly higher scores of 58.1 and 57.3, but with a much higher latency of 196.6 seconds. This showcase ReFeed’s practical efficiency and applicability considering that ReFeed performs almost as well as the much larger teacher model, while being approximately 4× faster.
>
> We hope this clarification helps distinguish the scope and intent of our work.
>
>
> - **Reference in Response (1/2) and Response (2/2)**
>
> [1] Wan et al, ACUEval: Fine-grained Hallucination Evaluation and Correction for Abstractive Summarization, In ACL-findings, 2024.
>
> [2] Wadhwa et al, Learning to Refine with Fine-Grained Natural Language Feedback, In EMNLP-findings, 2024.
>
> [3] Wan et al, MAMM-Refine: A Recipe for Improving Faithfulness in Generation with Multi-Agent Collaboration, In NAACL, 2025.
>
> [4] Song et al, FineSurE: Fine-grained SummarizatioEvaluation using LLMs, In ACL, 2024.
>
> [5] Tang et al, MiniCheck: Efficient Fact-Checking of LLMs on Grounding Documents, In EMNLP, 2024.

---

> ### Author Response · Authors · 2025-06-07
> **Gentle reminder to review the response**
>
> Dear Reviewer iDgk
>
> We find your comments and feedback invaluable and have done our best to address them. Your suggestions have been incorporated into our revised manuscript. As the rebuttal period draws to a close, we kindly request your review of our response. Please do not hesitate to contact us if you need any further clarification or have additional queries.
>
> Sincerely, The authors

---

> > ### Comment · Reviewer_iDgk · 2025-06-09
> >
> > Hi,
> >
> > Sorry for taking a while to respond. I appreciate the detailed author response. I also took the chance to go through all the reviewers' comments, and went through the paper again.
> >
> > Thank you for explaining the differences in your results with those in the orig UniSummEval paper. I agree that human evaluations are hard to reproduce, and appreciate the effort you took to share the results in Claude 2.1. The explanation and results definitely helped me better understand the impact of your proposal. I also appreciate the explanation on pipelines P1 to P4.
> >
> > All these inputs helped me better understand the paper. I will be revising my score upwards as well as correct parts of my review. Thanks!

---

> > > ### Author Response · Authors · 2025-06-10
> > > **Reply to the Reviewer iDgk**
> > >
> > > Thank you for adjusting the score. We sincerely appreciate your thoughtful comments.
> > >
> > > We will reflect your comments in the revised version of the paper.

---

### Official Review · Reviewer_rriG · 2025-05-13

**Rating:** 8
**Confidence:** 4
**Ethics Flag:** 1

**Summary:**

The paper provides a new framework refeed for text/summary refinement, incorporating multiple dimensions of text assessment. The paper provides abalation of variation in the proposed techniques along with a fine-tuning dataset.

**Questions To Authors:**

-  I think there is potential to extend this paper by incorporating more feedback dimensions like style, tense. I think this could have made the paper stronger

**Reasons To Accept:**

- The paper is well written and easy to follow. The experiments are sound and all the results are well backed by numbers.
- Novel approach to incorporate Multi-Dimensional assessment.
- The approach incorporates reflective reasoning and backtracking during CoT.
- The experiments and the comparison cover most of the variation that can be thought of in this setting.
- The final proposed approach is justified with experiments and baselines
- ReFeed performs better than the existing text refinement framework while improving the generation on multiple dimensions.
- They do test low- vs. high-quality feedback, but not feedback format variation or dropouts.

**Reasons To Reject:**

Important reasons:
- Potential comparison bias in fine-tuning evaluation (Section 5.6) comparing existing methodologies, DCR, and ACUEval.
   - The ReFeed model is fine-tuned using its reflective training data (SumFeed-CoT), whereas baselines like DCR and ACUEval are evaluated using this same ReFeed-trained model and not their respective methodologies. This creates a potential bias in ReFeed’s favor and raises questions about the fairness of the comparison.

Not a big concern:
- ReFeed Requires Teacher Distillation Infrastructure, this is compute intensive
- No Direct Evaluation on Generalization to Unseen Feedback Styles
   - Only one dimension’s feedback is available
   - Different feedback formats are used (e.g., scalar scores or freeform text)

Nitpicking:
- Fine-tuned model comparisons are not done for P1–P3 Baselines

---

> ### Author Response · Authors · 2025-06-03
> **Response to Reviewer rriG (1/2)**
>
> We appreciate your valuable comments and suggestions. We hope that the concerns can be resolved through our clarifications in this response.
>
> ---
> > **Weakness 1.** *Potential comparison bias in fine-tuning evaluation (Section 5.6) comparing existing methodologies, DCR, and ACUEval. The ReFeed model is fine-tuned using its reflective training data (SumFeed-CoT), whereas baselines like DCR and ACUEval are evaluated using this same ReFeed-trained model and not their respective methodologies. This creates a potential bias in ReFeed’s favor and raises questions about the fairness of the comparison.*
>
> We acknowledge the concern regarding potential bias in the evaluation. We believe we took steps to reasonably mitigate such bias through the following measures:
>
> * **GPT-4o (or Human) based Evaluation:** All experimental results were evaluated either by an external model, GPT-4o (not the ReFeed-trained model), or by human annotators. For automated assessment, we adopted FineSuRE, the latest protocol for summary evaluation. These evaluation methods do not introduce bias favoring our method over others, such as DCR and ACUEval, as they rely on a neutral model and blinded human annotators, eliminating advantages from familiarity or model-specific alignment.
>
> * **Backbone Consistency:** We use the exact same Llama3 series as the backbone for all baselines, including DCR and ACUEval, to ensure architectural consistency across methods. This choice helps minimize variability or unintended bias that could arise from differences in base model capacity, pretraining data, or underlying implementation, thereby enabling a more reliable and fair comparison.
>
> * **Use of Official Pre-trained Models:** We used the officially released fine-tuned checkpoint for DCR and followed the original prompting strategy for ACUEval, a prompt-based baseline. Training these baselines on our own data was infeasible, as they do not support reflective reasoning or the modular structure of ReFeed, including multi-dimensional feedback.
>
> ---
>
> > **Weakness 2.** *ReFeed Requires Teacher Distillation Infrastructure, this is compute intensive.*
>
> As described in Lines 221–230, we curated a high-quality dataset through careful data selection, allowing us to achieve significant performance gains with just 8K examples. While distillation does introduce some computational cost, ReFeed operates with modest requirements: training on this 8K dataset takes approximately two hours on 4×L40S GPUs using DeepSpeed and LoRA. While one might question whether SFT requires a large amount of data and extended training time, recent trends suggest otherwise. It has been increasingly demonstrated that a small amount of high-quality reasoning data can yield highly effective models, significantly reducing both data and computational requirements [1,2].
>
> [1] LIMO: Less is More for Reasoning, arxiv preprint arxiv:2502.03387, 2025.
>
> [2] S1: Simple test-time scaling,  arXiv preprint arXiv:2501.19393, 2025.

---

> ### Author Response · Authors · 2025-06-03
> **Response to Reviewer rriG (2/2)**
>
> > **Weakness 3.** *No Direct Evaluation on Generalization to Unseen Feedback Styles. (3-1) Only one dimension’s feedback is available; and (3-2) Different feedback formats are used (e.g., scalar scores or freeform text.*
>
> Thank you for your valuable comments. We offer additional clarification regarding: (3-1) using only single-dimensional feedback for ReFeed, and (3-2) the impact of different feedback styles.
>
> **(3-1): When using only single-dimensional feedback for ReFeed:** The table below shows the performance of ReFeed when only faithfulness feedback is provided (`ReFeed-Faith`). Compared to `P1-Faith`, a single-dimension pipeline using "receptive" reasoning (as described in Lines 150–151), `ReFeed-Faith` achieves greater gains in Faithfulness (+6.9 vs. +2.7). That is, reflective reasoning is beneficial even when only a single feedback dimension is available, outperforming the receptive approach.
>
> Nevertheless, using only single-dimensional feedback overlooks multi-dimensional alignment, resulting in lower average summary quality due to the alignment tax. `ReFeed` with multi-dimensional feedback achieves the best overall refinement.
>
>
> | Pipeline         | Faithfulness | Completeness | Conciseness | Average |
> |----------------|--------------|--------------|-------------|---------|
> | `Before Refine`  | 78.0         | 46.4         | 76.4        | 66.9    |
> | `P1-Faith`       | 80.7 (+2.7)  | 45.9 (-0.5)  | 80.4 (+4.0) | 69.0 (+2.1) |
> | `ReFeed-Faith` `  | 84.9 (+6.9)  | 47.0 (+0.6)  | 80.2 (+3.8) | 70.7 (+3.8) |
> | `ReFeed`        | 82.7 (+4.7)  | 60.0 (+13.6) | 83.4 (+7.0) | 75.4 (+8.5) |
>
>
> **(3-2): When different feedback formats are used:** Since ReFeed is trained on fixed-form textual feedback, it may exhibit limited robustness to freeform feedback. This is a common limitation of SFT, which tends to reflect the structure of training data. To further strengthen generalization, we plan to augment training with diverse freeform feedback and explore alternative training strategies such as RL-based ones. Thanks for your valuable comment for our future work.
>
> ---
>
> > **Weakness 4.** *Fine-tuned model comparisons are not done for P1–P3 Baselines.*
>
> Since P1–P3 showed limited performance due to trade-off and order bias, We only fine-tuned the strongest baseline P4. However, we agree that including fine-tuning results for P1–P3 would provide a more comprehensive evaluation. We will add these comparisons in the revision.
>
> ---
>
> > **Question 1.** *I think there is potential to extend this paper by incorporating more feedback dimensions like style, tense. I think this could have made the paper stronger.*
>
> Thank you for the suggestion. We acknowledge that incorporating dimensions like style and tense could improve the expressiveness of our pipeline. We plan to investigate such extensions in future work to offer more personalization.
>
> ---

---

> ### Author Response · Authors · 2025-06-07
> **Gentle reminder to review the response**
>
> Dear Reviewer rriG
>
> We find your comments and feedback invaluable and have done our best to address them. Your suggestions have been incorporated into our revised manuscript. As the rebuttal period draws to a close, we kindly request your review of our response. Please do not hesitate to contact us if you need any further clarification or have additional queries.
>
> Sincerely, The authors

---

> ### Comment · Reviewer_rriG · 2025-06-11
>
> Thanks for the response. I have went over the response. While I am satisfied with the response, it dose not answer the main concern I have raised. However I should also note that there is no practical way to address it effectively. I don’t want to change my scores for the paper.

---

### Author Response · Authors · 2025-06-11
**Summary of Author-Reviewer Rebuttal.**

We deeply appreciate the three highly expert reviewers' time and effort spent evaluating our paper. We are thankful for their insightful and constructive comments, which greatly contributed to refining our work. Here's a summary of the rebuttal:

There were several requests for clarification on our experiments from the reviewers. In response, we have taken the following actions:

For Reviewer rriG
- We provide a clearer explanation of the steps taken to mitigate potential bias issues in the evaluation process for fair comparisons.
- We discussed the potential performance issues including computational cost and the generalizability of the feedback format.

For Reviewer iDgk
- We conduct additional human evaluation using the same AI assistance model as in UniSumEval to provide an in-depth analysis of the performance discrepancy between UniSumEval.
- We provided clear clarifications on the motivations of reflective reasoning and necessity behind including P1 and P4 baselines that the reviewer inquired about.

For Reviewer qsDy
- We showed direct refinement model trained on refined summaries in SumFeed-CoT is less effective at addressing multiple dimensions compared to refinement based on reflective reasoning on multiple feedback.
- We presented the results using prompt engineering without distillation to confirm the importance of model alignment through distillation with Long-CoT for effective reflective reasoning.

For Reviewer 7Nyd
- We clarified our results based on automatic metrics are consistent with human evaluations.
- We conducted an ablation study to evaluate the contribution of individual substeps in our data construction pipeline, with a focus on diversified feedback and quality filtering.
- We discussed that the necessity of not controlling variables such as summary length.
- We provided a clear explanation of the rationale behind the filtering thresholds.
- We conducted prompt optimization for other baselines highlighting the necessity of reflective reasoning as employed in our Refeed.

We believe these responses effectively address any points of confusion and resolve all concerns raised by the reviewers. We will incorporate the reviewers’ suggestions in our revision.

Once again, we sincerely thank the reviewers for their time, positive and invaluable feedback.

Sincerely, the authors

---

### Decision · Program_Chairs · 2025-07-08

**Decision:**

Accept

**Comment:**

The paper provides a new framework for refining summaries based on reflective reasoning. The authors release a large-scale dataset containing chains of thought reasoning about summaries in terms of faithfulness, completeness and conciseness. They then trained smaller models on this dataset (distillation) to perform reflective reasoning and refine a summary. The experiments investigated several pipeline variants and assessed their effect on qualities such as faithfulness, completeness and conciseness.

Reviewers agree on a number of strengths, which make this paper a worthwhile contribution to the conference. The paper is well written and focuses on important challenges in refining summaries along multiple dimensions. The proposed approach to refining summaries along these multiple aspects is original and is shown to outperform existing text refinement frameworks and baselines, particularly in trading-off multiple quality dimensions. The experimental analysis is thorough, considering important variations of the method and building on recent datasets and metrics, and makes a strong case for the efficacy of the method. A new dataset is presented in the paper that could be useful for future research.

Some clarifications provided by the authors in the rebuttal phase would be valuable to include in the final version of the paper, if accepted, such as the motivations for targeting reflective reasoning and the shift from passively accepting feedback to reflecting on it. The human evaluation using Claude v2.1, along with the accompanying explanation, will be important for understanding differences in performance metrics to prior work. The results for the additional baseline ‘direct refinement’ provides an interesting point of comparison that further demonstrates the effectiveness of ReFeed. Likewise, the ablation results further support the complete approach and should be added to the camera-ready version.

Finally, please note some constructive suggestions from reviewers on presentation, which was generally very good, but could be improved in a couple of places, e.g., Figure 1.